# PepQuery2 democratizes public MS proteomics data for rapid peptide searching

Bo Wen [1,2,3] & Bing Zhang [1,2] ✉

We present PepQuery2, which leverages a new tandem mass spectrometry (MS/MS) data indexing approach to enable ultrafast, targeted identification of novel and known peptides in any local or publicly available MS proteomics datasets. The stand-alone version of PepQuery2 allows directly searching more than one billion indexed MS/MS spectra in the PepQueryDB or any public datasets from PRIDE, MassIVE, iProX, or jPOSTrepo, whereas the web version enables users to search datasets in PepQueryDB with a user-friendly interface. We demonstrate the utilities of PepQuery2 in a wide range of applications including detecting proteomic evidence for genomically predicted novel peptides, validating novel and known peptides identified using spectrum-centric database searching, prioritizing tumor-specific antigens, identifying missing proteins, and selecting proteotypic peptides for targeted proteomics experiments. By putting public MS proteomics data directly into the hands of scientists, PepQuery2 opens many new ways to transform these data into useful information for the broad research community.

Tandem mass spectrometry (MS/MS)-based shotgun proteomics is the workhorse for protein identification and quantification in biomedical research. Thousands of shotgun proteomics datasets with billions of MS/MS spectra have been generated and deposited into public data repositories such as PRIDE[1], MassIVE, iProX[2], jPOSTrepo[3], and Proteomics Data Commons (PDC). However, it remains challenging to put these public MS proteomics data directly into the hands of scientists to address their research questions, an important step to unleash the full potential of these data.

One way to democratize the use of these public data is to allow users to query peptide or protein sequences of interest against MS/MS spectra in a public data repository to identify high-quality peptide-spectrum matches (PSMs), similar to BLASTing a DNA sequence of interest against a genomic sequence database to identify sequences of high similarity. PSMs identified from the public MS/MS data may provide evidence to support novel peptide predictions[4–7], to prioritize putative tumor-specific antigens[8], to uncover "missing" proteins[9], among other applications[7,10]. Many tools, such as MaxQuant[11], MS-GF + [12], Comet[13], Open-pFind[14], and MSFragger[15], have been developed to search MS/MS data against a prespecified protein database to

identify PSMs, but the common goal of these tools is to comprehensively interpret all observed MS/MS spectra in a study. These spectrum-centric tools are not suitable for peptide-centric analysis that aims to identify one or more peptide or protein sequences of interest, because most of the computational time is spent on evaluating peptide-spectrum pairs irrelevant to the query sequences. Moreover, they typically lack rigorous quality control for individual PSMs.

To complement the spectrum-centric approaches, we have previously developed PepQuery, a peptide-centric search engine for MS/MS data analysis[16]. PepQuery allows users to query a novel sequence of interest against an MS/MS spectra database to identify statistically significant PSMs. By focusing on only the query sequence of interest, PepQuery bypasses all unnecessary computations, leading to vastly reduced space and time complexity. Moreover, due to increased time efficiency, PepQuery further enables the comprehensive examination of peptide modifications to reduce false discoveries. Most studies using the spectrum-centric database searching tools for peptide identification consider only a small number of protein modifications. As a consequence, false discoveries arise when a spectrum matched to one peptide has a better match to another peptide with a modification

[1]Lester and Sue Smith Breast Center, Baylor College of Medicine, Houston, TX 77030, USA. [2]Department of Molecular and Human Genetics, Baylor College of Medicine, Houston, TX 77030, USA. [3]Present address: Department of Genome Sciences, University of Washington, Seattle, WA 98195, USA. ✉e-mail: bing.zhang@bcm.edu

not considered in data analysis. This problem is well recognized in novel peptide identification[17], and it is also common in normal peptide identification[15]. Comprehensively considering all types of natural and artificial protein modifications, PepQuery has been shown to be highly effective in reducing false discoveries in novel peptide identification[16,18]. PepQuery was implemented both as a stand-alone tool and a web application. The stand-alone version supports the analysis of local proteomics datasets. The web version enables web-based analysis of public proteomic datasets available in PepQueryDB, but the analysis is limited to one query sequence and one dataset at a time, and there is no support for direct analysis of MS/MS data in the major proteomics data repositories.

Here we introduce PepQuery2, which leverages a new MS/MS indexing approach and cloud storage to enable ultrafast, targeted identification of both novel and known peptides. The stand-alone version of PepQuery2 allows users to search more than one billion MS/MS data indexed in the PepQueryDB from any computers with internet access. It also supports direct analysis of user-provided MS/MS data, any public datasets in PRIDE, MassIVE, iProX, or jPOSTrepo, or Universal Spectrum Identifiers (USIs)[19] from ProteomeXchange (Fig. 1). Meanwhile, we have extended the web version to include public proteomics datasets from all flagship CPTAC studies, leading to a total of 48 datasets. We demonstrate the utilities of PepQuery2 in detecting proteomic evidence for genomically predicted novel peptides, validating novel or known peptides identified from spectrum-centric database searching, prioritizing tumor-specific antigens, identifying missing proteins, and selecting proteotypic peptides for targeted proteomics experiments.

## Results

### An overview of PepQuery2

One of the fundamental improvements in PepQuery2 is the new MS/MS spectrum indexing by leveraging the cloud storage service (Methods, Supplementary Fig. 1). In brief, MS/MS spectra from a dataset with similar precursor mass after mass binning are stored in a single compressed file on the cloud storage so that candidate spectra matched to a query peptide can be retrieved quickly from large-scale datasets. This new indexing method not only enables ultrafast peptide-spectrum matching but also greatly reduces data storage space compared with the previous SQL-based indexing. More than one billion MS/MS spectra have been indexed and are available in the PepQueryDB, which includes data from 48 large-scale human proteomics datasets of more than 1500 tumors covering 14 types of cancers, 375 cell lines, and more than 30 types of normal tissues (Fig. 1, Supplementary Data 1).

Another major improvement in PepQuery2 enables using known peptide or protein sequences, in addition to novel sequences, as a query sequence (Fig. 1). This is particularly useful for hunting for missing proteins, validating unexpected but interesting identifications, such as tumor-associated antigens, selecting proteotypic peptides for targeted proteomics experiments, and validating peptides identified from small-scale MS experiments, such as affinity purification–mass spectrometry (AP-MS) experiments, in which quality control of protein identification remains challenging. Moreover, in addition to peptide validation, PepQuery2 has been expanded to support PSM validation. This new feature allows users to use the peptide-centric analysis as a complementary approach to validate interesting PSMs identified in spectrum-centric analysis and classify them into seven different categories (c1-c7, Fig. 1, Methods). C1 (exact ref match) includes input PSMs for which the peptide has an exact match to a sequence in the reference database. This is only applicable to novel peptide validation, and PSMs in this category are essentially invalid novel identifications. This could happen when the input PSMs are identified using a different database than the selected reference database in PepQuery2 analysis. C2 (no candidate spectrum) includes input PSMs for which the peptide has no candidate spectrum based on the peptide mass and allowed mass error tolerance. C3 (low score) includes input PSMs with a PepQuery2 computed PSM score lower than a prespecified threshold (Methods). These low-quality matches are excluded from further analysis to save time. C4 (equal or better ref match) includes input PSMs for which the spectrum can be matched to a reference peptide with an equal or better PSM score. C5 (insignificant score) includes input PSMs failing the statistical evaluation based on randomly shuffled peptides (Methods). C6 (better mod ref match) includes input PSMs for which the spectrum can be better matched to a reference peptide with a modification that is not considered in the spectrum-centric analysis. Input PSMs passing all these filtering steps are considered as C7, or confident identifications.

### Detecting proteomic evidence for genomically predicted novel peptides

To demonstrate the utility of PepQuery2 in identifying proteomic evidence from public data to support genomically predicted novel peptides, we queried the KRAS G12D peptide (LVVVGA**D**GVGK) against 12 CPTAC global proteomics datasets indexed in PepQueryDB, including 210,282,541 MS/MS spectra from one label-free experiment and 11 tandem mass tag (TMT) experiments spanning 10 cancer types (Methods). We identified 28 PSMs from five cancer types in which *KRAS* G12D somatic mutation is frequently reported (Fig. 2a, Supplementary Data 2), and a representative MS/MS spectrum identifying the mutant peptide in a pancreatic cancer sample is shown in Fig. 2b. Among the 28 PSMs, 22 were from samples with corresponding genomic mutation, two were from samples without genomic sequencing data, three were from samples with a *KRAS* G13D mutation, and one was from a sample with an *HRAS* G12D mutation. Of note, the HRAS G12D peptide is the same as the KRAS G12D peptide, and the KRAS G13D peptide (LVVVGA**GD**VGK) is hard to distinguish from the KRAS G12D peptide (LVVVGA**DG**VGK) based on MS/MS spectrum since there is only a minor difference between the two sequences. These data suggest high specificity of the PepQuery2 analysis despite the huge search space, including >210 million MS/MS spectra. Meanwhile, among the 75 tumor samples with the *KRAS* G12D mutation detected at DNA level, 41 (55%) had protein evidence from the PepQuery2 analysis (Fig. 2a). The sensitivity of 55% (41/75) is outstanding considering the well-recognized low sensitivity of MS proteomics in detecting mutant peptide[20]. Importantly, the whole analysis took less than four minutes on a Linux server with 40 threads, or five minutes on a Mac computer using eight threads. With the spectrum-centric approach, the same task would take several days for data downloading, customized database preparation, and database searching.

### Validating novel peptide identifications

Another utility of PepQuery2 is to validate novel peptides identified in customized database-based spectrum-centric analysis, in which false discovery is a major concern[17,18]. We illustrated this feature by validating novel peptides resulted from tryptophan-to-phenylalanine codon reassignment (W > F), which has recently been reported to occur frequently in human cancer[21]. In the original study[21], a total of 473 peptides with W > F substitution were matched to 3011 spectra from the CPTAC lung squamous cell carcinoma (LSCC) dataset[22] using a customized database searching strategy. Strikingly, only 9.2% of the PSMs reported in the original study passed PepQuery2 validation (C7 group in Fig. 3a, Supplementary Data 3), corresponding to a 13.5% (64 out of 473) validation rate at the peptide level. The group of PSMs passing PepQuery2 validation (C7) had significantly higher PSM scores (i.e., PeptideProphet probability scores) in the original study compared with the other groups except for C1 and C2 (Fig. 3b). Peptides in the C1 group had exact matches to sequences in the reference database used in the PepQuery2 analysis. These are high-quality identifications as indicated by excellent PeptideProphet probability scores,

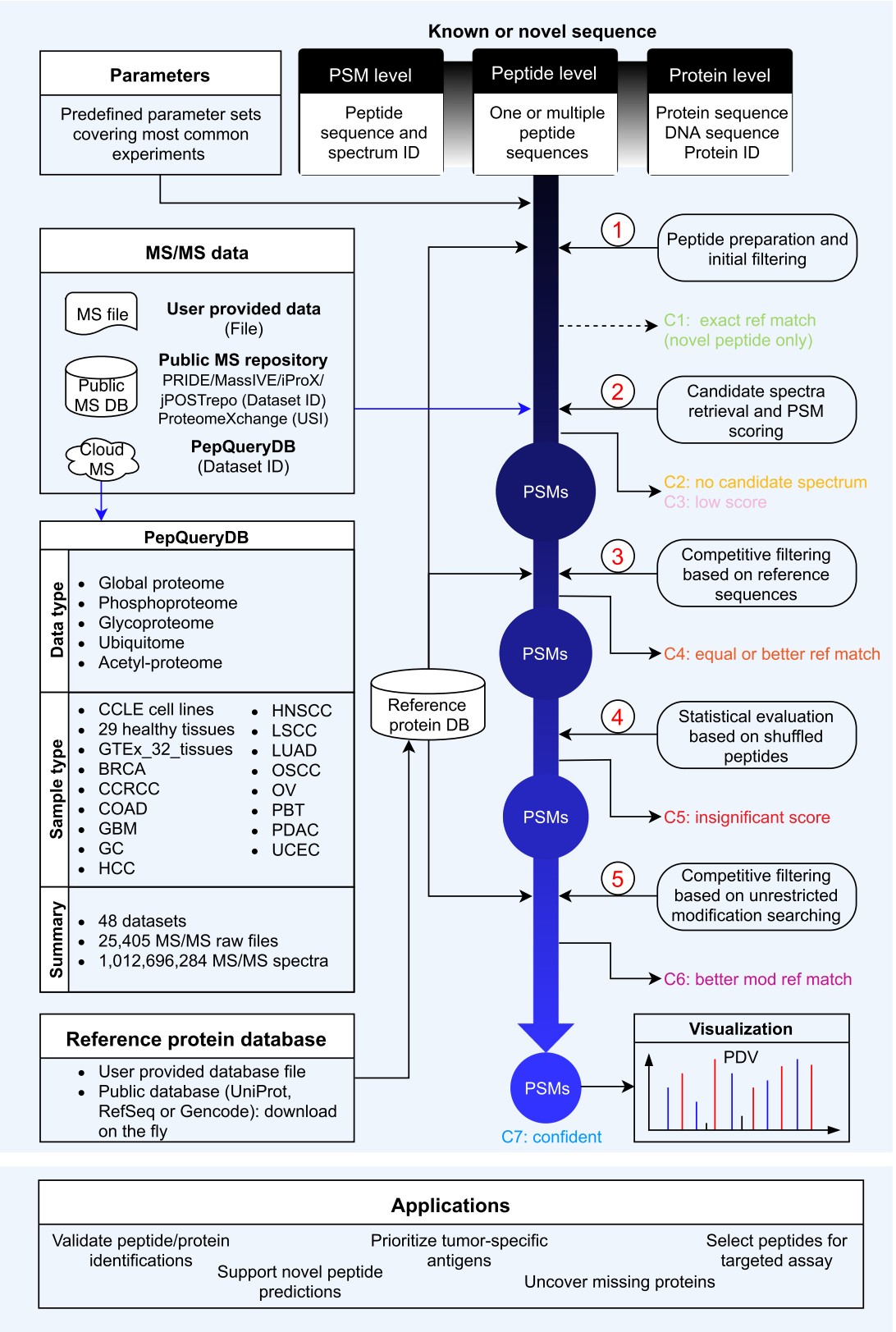

**Fig. 1 | Overview of the PepQueryDB, PepQuery2 workflow, and some possible applications.** The workflow involves five major steps as described in the previous publication (ref. 16): (1) peptide preparation and initial filtering; (2) candidate spectra retrieval and PSM scoring; (3) competitive filtering based on reference sequences; (4) statistical evaluation based on shuffled peptides; and (5) competitive filtering based on unrestricted modification searching. Query peptides and PSMs are classified into seven categories (C1-C7) on the basis of query results.

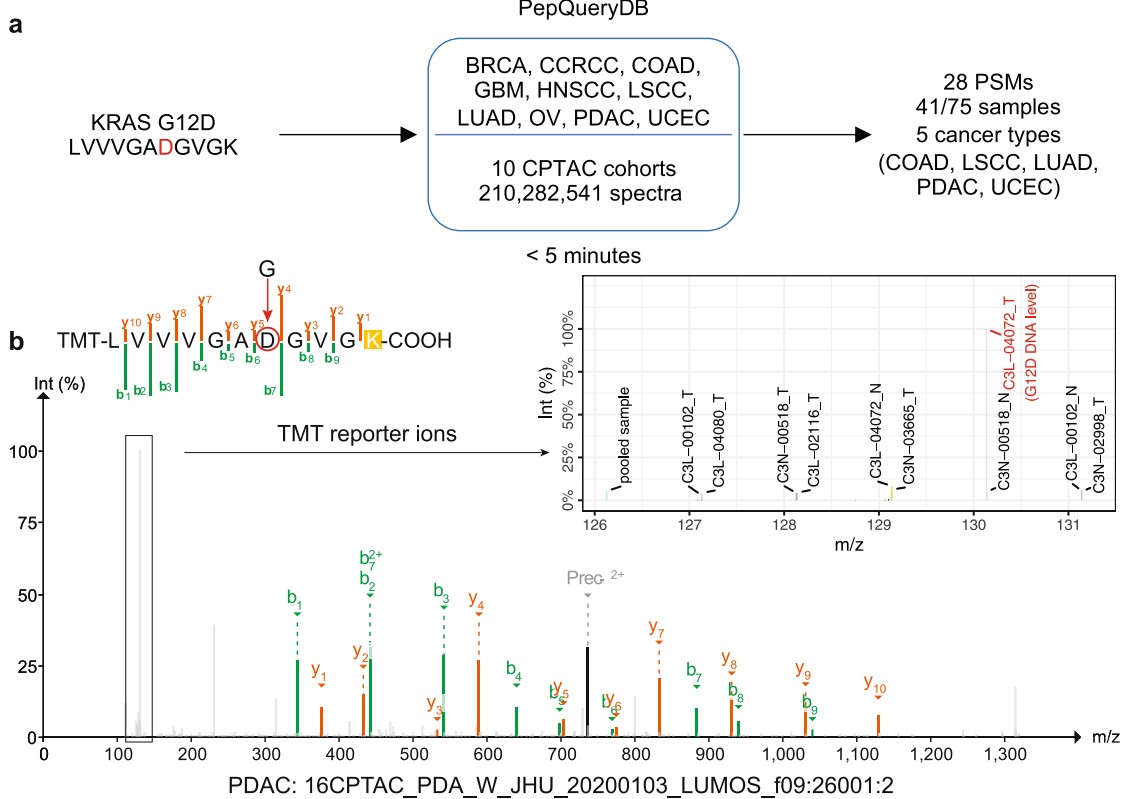

**Fig. 2 | KRAS G12D variant peptide identification. a** Identification of a novel peptide resulted from the *KRAS* G12D mutation in 10 CPTAC cohorts. **b** A representative MS/MS spectrum identifying the mutant peptide in a pancreatic cancer sample. The TMT reporter ion of the sample with *KRAS* G12D mutation evidence at DNA level (highlighted in red) shows much higher intensity than those from samples without this mutation. Source data of 2a are provided as a Source Data file.

but they do not qualify as novel peptides because they can be mapped to reference sequences in GENCODE. The C2 group included only 0.2% of the PSMs for which PepQuery2 failed to retrieve candidate spectra based on the pre-specified mass error tolerance. The C3 and C5 groups accounted for 11% of the previously reported PSMs, and they failed PepQuery2 validation due to low and insignificant PSM scores, respectively (Methods). The C4 group, in which the spectra had better matches to a reference peptide when considering both fixed and variable modifications specified in the original search, included 36.9% of the PSMs, and for an additional 36.7% (C6 group), the spectra had better matches to a reference peptide with modifications that were not considered in the original database searching. Within the C4 group, 48% had better matches to a reference peptide containing an amino acid W (Fig. 3c-d, Supplementary Fig. 2), and these potential false identifications from the original study can be attributed to the lack of competition from peptides containing amino acid W because in the customized database used in the original study, all Ws were replaced with an F without keeping a version of the unaltered sequence. Using Delta RT, the absolute difference between observed retention time (RT) and deep learning-predicted RT of the same peptide, as an additional evaluation metric[18], we further showed that the PSMs passing PepQuery2 validation had significantly lower delta RT compared with those failing validation (Fig. 3e). Repeating the spectrum-centric analysis using a customized database including human reference proteins downloaded from UniProt (Methods) reduced the total number of candidate peptides with W > F substitution from 473 to 240 (1024 PSMs) and the proportion of the C4 group in PepQuery2 validation to 27% (Supplementary Fig. 2b). Despite these improvements, there were still many candidate PSMs failing PepQuery2 validation, including 510 PSMs (50%) for which the spectrum was matched to a reference peptide with a modification not considered in the spectrum-centric analysis (C6). Together, these results demonstrate PepQuery2 as an effective tool for identifying mistakes in customized database construction (a task not required for PepQuery2 analysis) as well as potential false positives among the novel peptides identified in spectrum-centric analysis.

## Validating known peptide identifications

False discoveries are not unique to novel peptide identifications. For example, validation of known peptide identifications using alternative computational algorithms is very useful for small-scale MS experiments, such as AP-MS. Due to limited number of proteins present in an MS sample in these experiments, the normally used target-decoy-based strategy[23] may not be able to provide accurate false discovery rate (FDR) estimation. The peptide-centric analysis in PepQuery2 does not rely on the traditional target-decoy strategy used for global FDR estimation and thus provides a statistically unrelated method for validating spectrum-centric analysis results. With the new improvement that allows using known peptide or protein sequences as a query sequence, such validation is now possible in PepQuery2. To illustrate this utility, we applied PepQuery2 to validate prey proteins identified when HDAC1 was used as the bait in AP-MS experiments performed on the 293 T cell line in the BioPlex 3.0 project[24]. Using PepQuery2 to query the 86 prey proteins identified in the original study against MS/MS data available in MassIVE, 85 passed the validation but one failed. CHD5 is the prey protein failing PepQuery2 validation, and the original identification was based on one unique CHD5 peptide. All spectra associated with the originally reported PSMs identifying this peptide had equal or better match to a peptide from CHD4 with deamination (Fig. 4a, Supplementary Fig. 3), a modification not considered in the original analysis. CHD4, but not CHD5, is a known component in HDAC1-containing protein complexes in the CORUM database, such as the nucleosome remodeling and deacetylation complex (Fig. 4b), and 80 spectra from the same AP-MS experiments were matched to 25

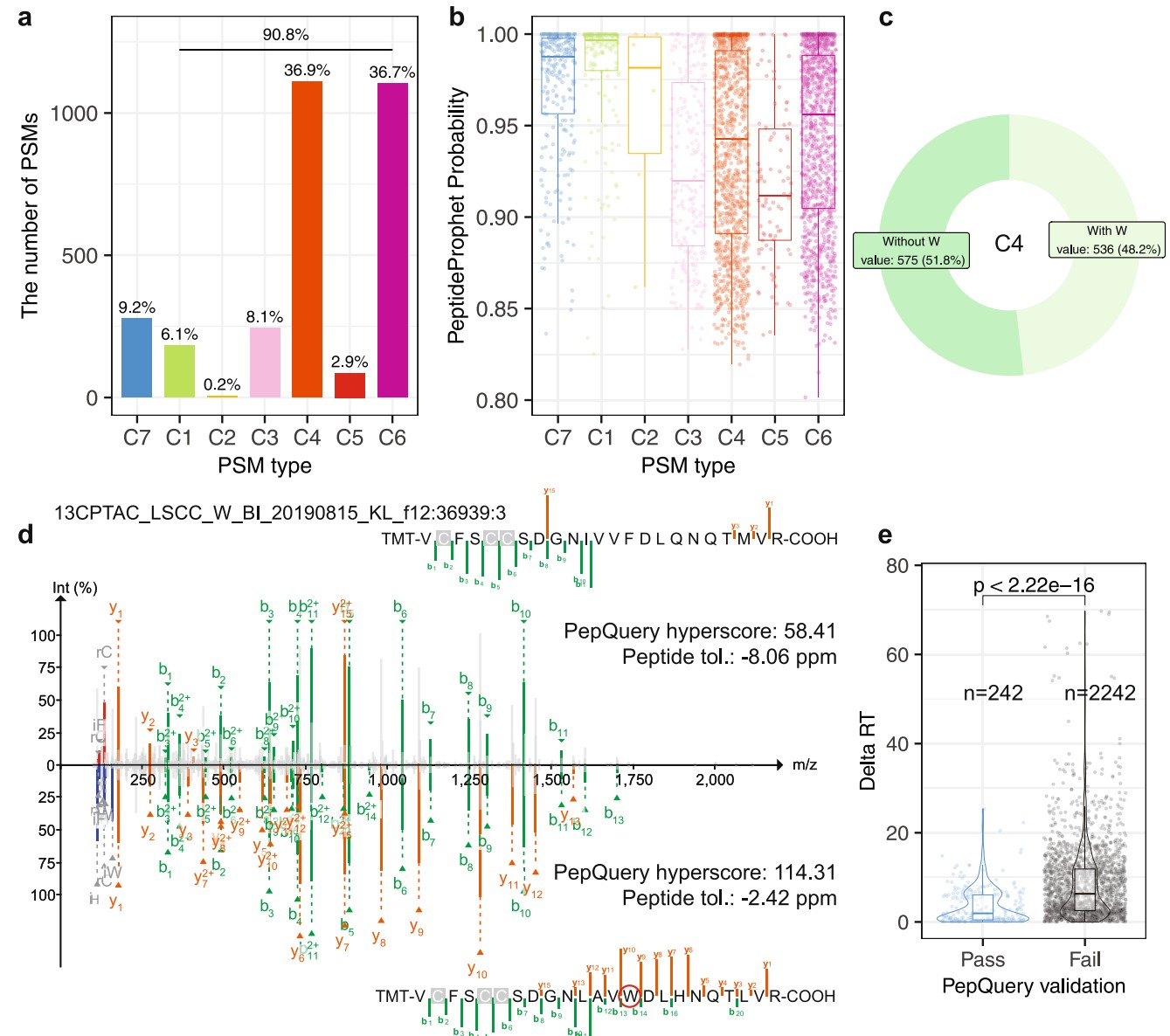

**Fig. 3 | Validation of novel peptide identifications. a** PepQuery2 classified previously reported PSMs supporting novel peptides resulted from W to F substitution into seven categories as described in Fig. 1, and only C7 PSMs passed the validation. **b** PeptideProphet probability distributions for different categories of the PSMs. **c** The percentages of PSMs (**C4**) from W2F peptides in which the spectrum matched equally or better to a reference peptide with or without containing amino acid W. **d** The spectrum originally matched to a W2F peptide has better match to a reference peptide containing amino acid W in PepQuery2 validation. **e** Delta RT distributions for PSMs passing or failing PepQuery2 validation. The *p*-value was calculated using two-sided Wilcoxon rank sum test. For boxplots, centerline indicates the median, box limits indicate upper and lower quartiles, whiskers indicate the 1.5 interquartile range. Source data of 3a, 3b, 3c and 3e are provided as a Source Data file.

unique peptides from CHD4. These data suggest that CHD5 is likely to be a false identification in the original analysis. In addition to the HDAC1 experiment, AP-MS experiments performed on the 293 T cell line in the BioPlex 3.0 project for 11 other bait proteins identified CHD5 as a prey protein. Ten out of these 11 identifications failed PepQuery2 validation for the same reason.

Even in large-scale MS experiments where peptide identification FDR can be well controlled, further validation is still useful, especially for unexpected identifications with important biological or clinical implications. As an example, PGK2 was identified and quantified in the CPTAC lung adenocarcinoma (LUAD) global proteome dataset both in the original publication[25] using the Spectrum Mill search engine and in an independent analysis using FragPipe. *PGK2* was initially assumed to be a pseudogene but was later found to have highly specific expression

in testis[26]. The detection of PGK2 in the LUAD cohort was unexpected. Moreover, a peptide (AVVLMSHLGRPDGVPMPDK) supporting the detection of PGK2 was identified in all samples with high abundance (Fig. 4c), and the abundance in tumors was significantly higher than in adjacent normal tissues (*p* < 2.2e-16, two-sided Wilcoxon Rank Sum test, Fig. 4c-d), suggesting PGK2 as a possible cancer/testis (CT) antigen and a putative immunotherapy target for lung cancer treatment. To further investigate this intriguing finding with potential clinical significance, we used PepQuery2 to validate all 239 PSMs reported by FragPipe for PGK2. Remarkably, 94% (224) of the PSMs, including all 102 PSMs identifying the highly abundant peptide, failed the validation (Fig. 4c). The spectra matched to the peptide AVVLMSHLGRPDGVPMPDK by FragPipe were found by PepQuery2 to have a better match to another peptide SVVLMSHLGRPDGVPMPDK

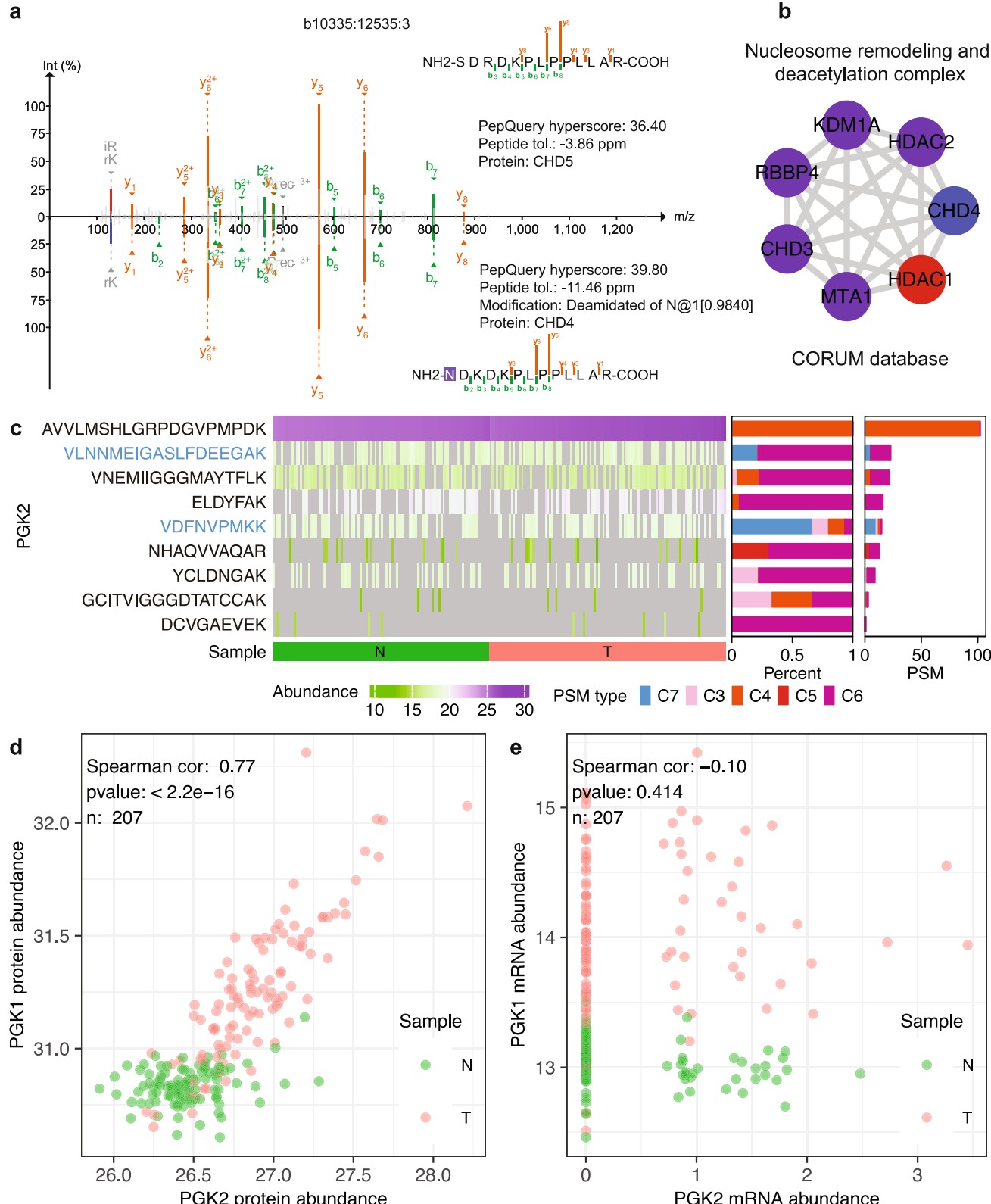

**Fig. 4 | Validation of known peptide identification. a** An originally reported spectrum identifying CHD5 in an AP-MS experiment using HDAC1 as the bait was found by PepQuery2 to have a better match to a peptide from CHD4 with deamidation. **b** A protein complex containing both HDAC1 and CHD4 in the CORUM database. **c** PepQuery2 classified previously identified and quantified PGK2 peptides into different categories. Only the peptides highlighted in blue font had one or more PSMs passing PepQuery2 validation. **d** Protein abundance correlation between PGK1 and PGK2. **e** mRNA abundance correlation between PGK1 and PGK2. For d and e, only samples with non-missing and non-zero values in both samples were considered when calculating the spearman correlation. Source data of 4c, 4d, and 4e are provided as a Source Data file.

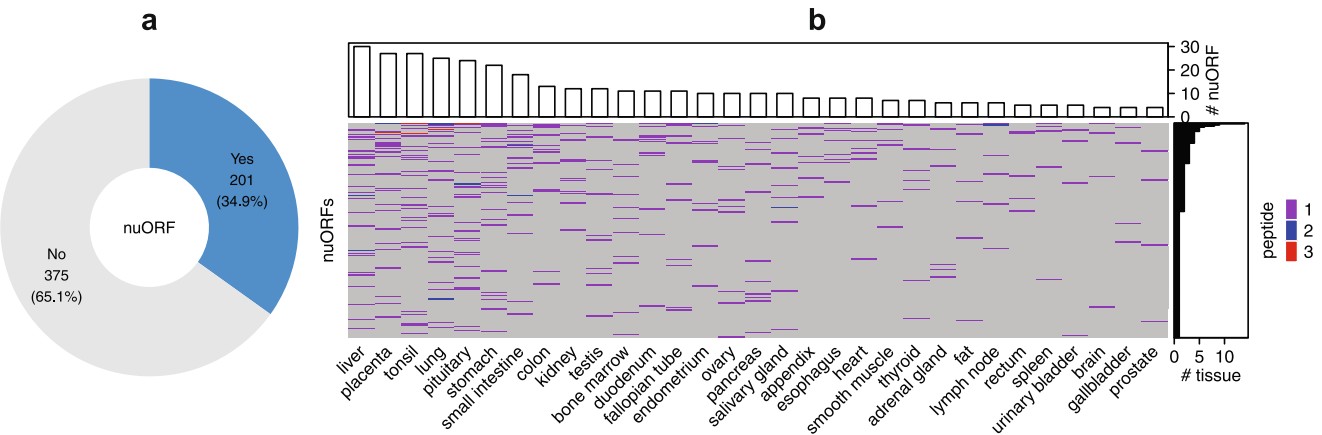

**Fig. 5 | Prioritization of tumor-specific antigens. a** Summary of identification results for querying 576 nuORFs against a public proteomics dataset on 31 healthy tissues. **b** The distribution of nuORF identifications across different healthy tissues. Source data are provided as a Source Data file.

from the homologous protein PGK1 (Supplementary Fig. 4). Further investigation showed that the PGK2 protein abundance quantified by FragPipe is highly correlated with the PGK1 protein abundance (Fig. 4d), whereas the *PGK2* mRNA abundance does not correlate with *PGK1* mRNA abundance in the same tumor cohort, and all tumors had very low *PGK2* mRNA abundance, including more than half with no *PGK2* mRNA detection (Fig. 4e). PGK2 identification was also reported in a colon cancer study[27], where the supporting PSMs also failed PepQuery2 validation because the associated spectra had an equal or better match to PGK1 peptides (Methods, Supplementary Fig. 5). PGK2 is not an exception. As another example, NAA11, a protein shown restricted expression in reproductive tissues in Human Protein Atlas, was also identified by FragPipe in the CPTAC LUAD dataset. All PSMs from this protein failed PepQuery2 validation because the associated spectra had equal or better match to a homologous peptide from NAA10, and correlation analyses between NAA10 and NAA11 at protein and mRNA levels provided further evidence to confirm the invalidity of the NAA11 identification and quantification (Supplementary Fig. 6). These results demonstrate the power of PepQuery2 as a validation tool to safeguard the quality of unexpected normal peptide identifications to prevent unnecessary downstream experimental and clinical investigations.

## Prioritizing tumor-specific antigens

In addition to validating peptide identifications using publicly available data from the same study, PepQuery2 can also be used to search interesting peptides identified from one study in other public datasets to gain new insights. A recent cancer immunopeptidomics study identified major histocompatibility complex (MHC) bound peptides from 576 novel or unannotated open reading frames (nuORFs) in cancer immunopeptidomics data[28]. These nuORFs greatly expand the cancer antigen repertoire and putative immunotherapy targets. To test their cancer specificity, we queried these nuORFs against a published heathy tissue dataset containing proteomics data (68,225,841 MS/MS spectra) from 31 heathy tissues[29]. Among the 576 nuORFs, 201 (35%) were detected in at least one and 83 (14%) in two or more healthy tissues (Fig. 5, Supplementary Data 4). Thus, PepQuery2 quickly narrowed down the list of nuORFs for further investigation as candidate neoantigens.

## Hunting for missing proteins

In the human proteome project (HPP), human proteins in the neXtProt database are classified into different categories based on the strength of supporting evidence[9]. A total of 1343 proteins are still classified as "missing proteins" due to the lack of experimental evidence at the protein level, and an important goal of the HPP is to hunt for the missing proteins in human proteome[9]. To look for MS-based experimental evidence for the 1343 missing proteins from public proteomics

data, we used PepQuery2 to query these proteins against all datasets in PepQueryDB, including >1 billion MS/MS spectra. Following the HPP guideline for missing protein identification[30], 48 missing proteins were identified with two or more unique peptides detected in at least one dataset, and all of them had peptide evidence in multiple datasets (Fig. 6, Supplementary Data 5, Methods). These proteins could serve as candidates for further validation using targeted proteomics with synthetic peptides or functional studies. This quick analysis brought the missing protein search a step closer to completion.

## Guiding the selection of proteotypic peptides for targeted proteomics

In targeted proteomics experiments, one of the critical steps is proteotypic peptide selection for the proteins of interest[31]. Analyzing public proteomics datasets could provide valuable empirical data to guide the selection of proteotypic peptides. Because public PTM proteomics data can also be included in the search, such analysis may also identify PTM peptides for targeted proteomics. As an example, we applied PepQuery2 to identify peptides specific to two protein isoforms of the gene *XBP1* (X-box binding protein 1), the conventionally spliced isoform XBP1u and the non-conventionally spliced isoform XBP1s. Importantly, unconventional splicing of *XBP1* mRNA by IRE1α is an indicator of unfolded protein response and plays an important role in several diseases, including cancer[32,33]. We searched the two protein isoforms against all the datasets available in PepQueryDB and detected four peptides from XBP1u, five peptides from XBP1s, and 19 peptides shared by the two isoforms (Fig. 7). Interestingly, all four peptides from XBP1u were detected from global proteome datasets, whereas four out of the five peptides from XBP1s were detected from phosphoproteome datasets. Thus, our analysis not only identified isoform-specific peptides but also showed that including phosphorylated peptides as targets may increase the chance of detecting XBP1s. In addition to guiding peptide selection, the identified spectra by directly searching MS/MS data also provide valuable information for transition list generation in multiple reaction monitoring (MRM)-based targeted assays. Since PepQuery2 not only enables searching more than one billion MS/MS spectra from a diverse collection of cancers, cell lines, and normal tissues in PepQueryDB but also allows searching other MS/MS data in public proteomics data repositories such as PRIDE, the large volume of MS/MS data in these repositories could be readily used to guide targeted experiment design for proteins of interest.

## Discussion

One of the most important milestones in proteomics is the Amsterdam Principles[34], which require mandatory raw MS/MS data deposition to promote broad reuse of the data. However, because of the challenges

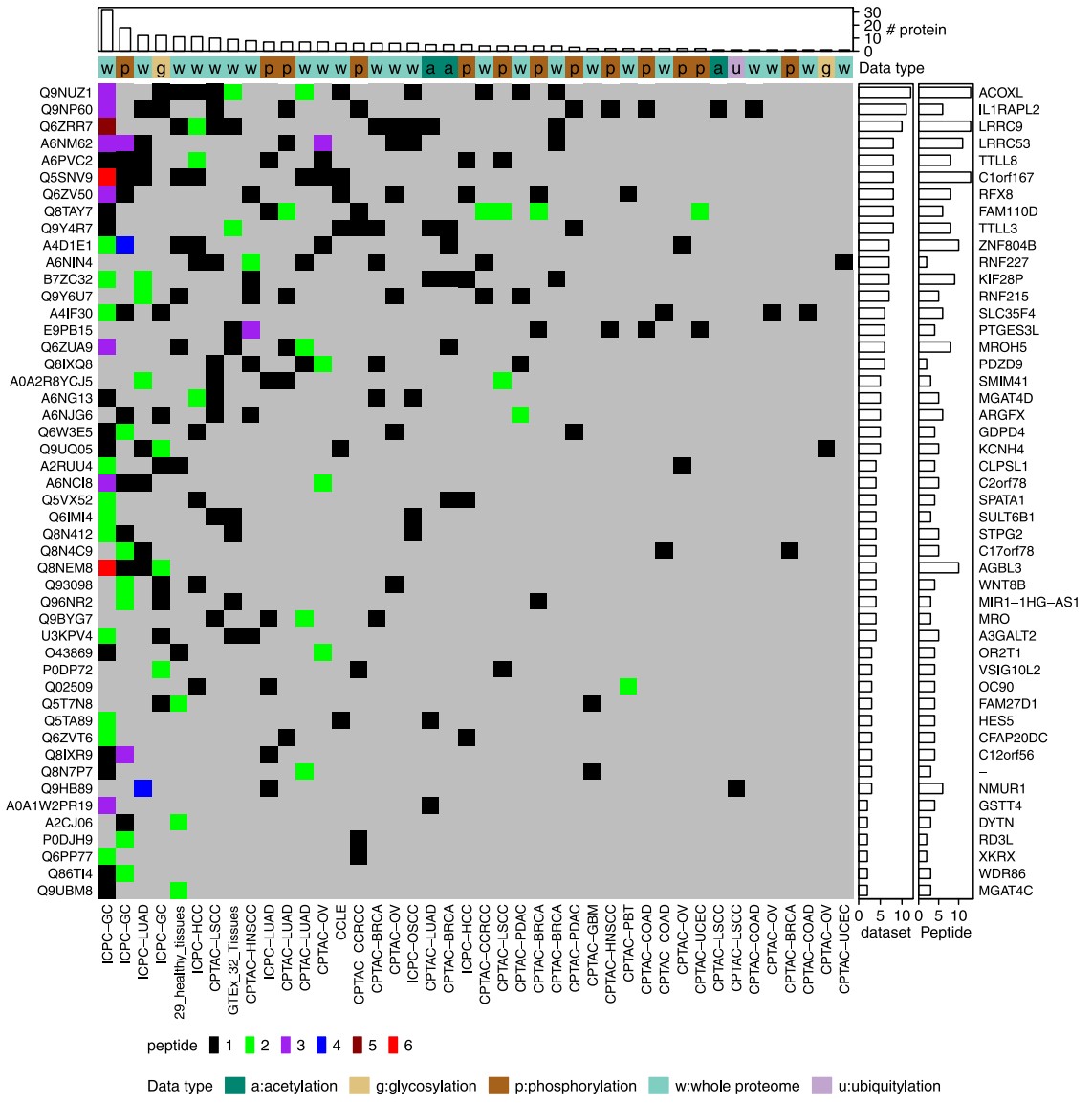

**Fig. 6 | Missing protein identification.** The distribution of missing protein identifications across different MS/MS datasets in PepQueryDB. Source data are provided as a Source Data file.

involved in understanding, downloading, analyzing, and interpreting MS/MS data, investigation and reuse of these public data are largely restricted to computational proteomics researchers. By enabling rapid identification of any known or novel peptide sequences of interest in any local or publicly available MS-based proteomics datasets in a targeted manner, PepQuery2 provides a practical solution that makes public MS/MS data easily useful to the general research community.

Recently, there is an increasing trend of reusing public proteomics data, especially in the context of searching for proteomic evidence of novel peptides and proteins with important biological and clinical implications. Unfortunately, usage complexity and often-overlooked inherent limitations of the spectrum-centric database searching tools may lead to false discoveries in such studies, as exemplified in the recently reported identification of W > F substitutants from previously published CPTAC MS/MS data[21] (Fig. 3). We further showed that false discoveries are also common in the identification of known proteins (Fig. 4). Our peptide-centric analysis complements the spectrum-centric database searching algorithms and provides an efficient framework for validation of important findings.

Both PepQuery and PepQuery2 use stringent criteria for novel peptide validation. When a candidate novel peptide and a reference

peptide have equal matching score to a spectrum, the spectrum is preferentially associated with the reference peptide based on the consideration that the prior probability of observing a novel peptide is much lower than a reference peptide. The competitive filtering step based on unrestricted modification searching further excludes the possibility that the spectrum has a better match to a reference peptide sequence with a modification not considered in the spectrum-centric analysis. When two peptides have equal or close scores to the same spectrum, it is very useful to manually check the matches. The web server of PepQuery2 provides annotated spectra for visual check of the matches. For the standalone version, details required for manual checking can be exported for visualization using PDV[35]. In this study, we also identified independent evidence to support PepQuery2 results. For the W2F peptide validation, we used a metric derived from retention time prediction to evaluate the quality of the results. For the cases of PGK2 and NAA11, we made use of paired mRNA data to support PepQuery2 results. For CHD5, we used prior knowledge about protein complex. Such analyses are very helpful in evaluating matches with equal or similar scores.

The new MS/MS data indexing method implemented in PepQuery2 enables retrieving candidate spectra from a large-scale

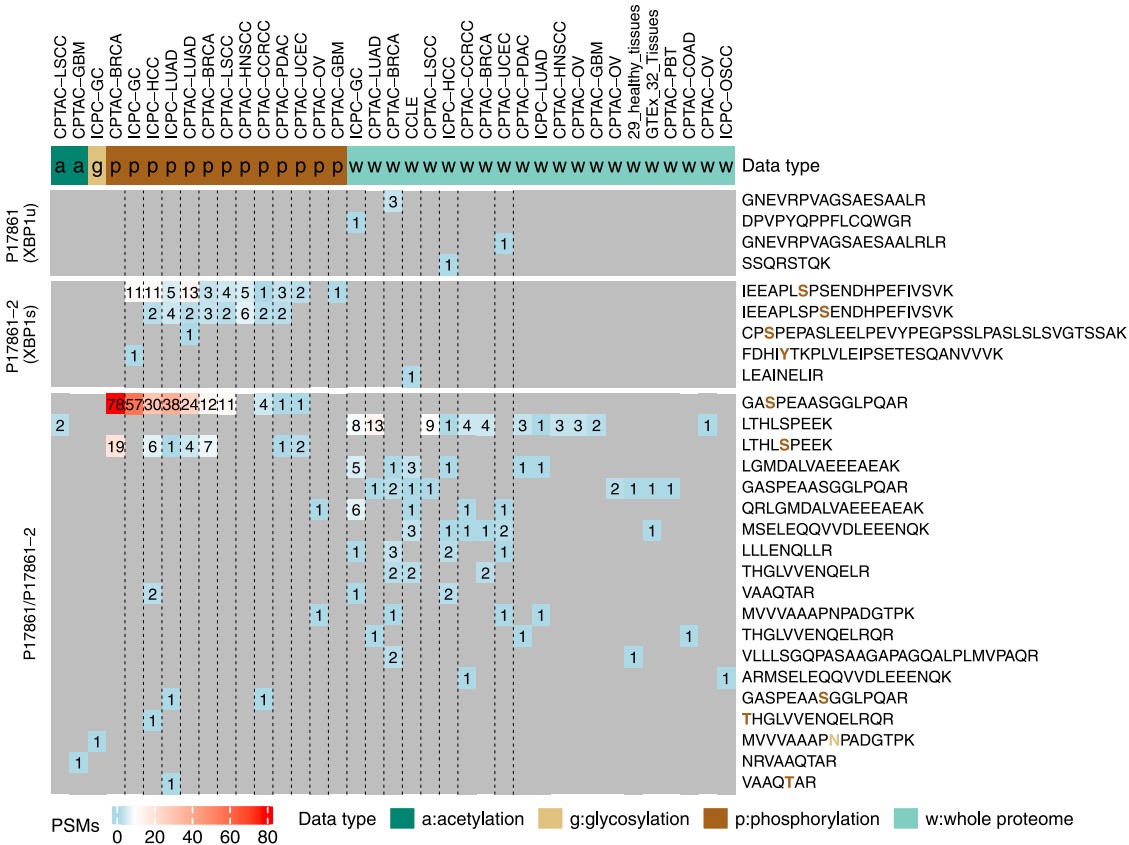

**Fig. 7 | Protein isoform identification.** The distribution of XBP1 isoform-specific and shared peptide identifications across different MS/MS datasets in PepQueryDB. Source data are provided as a Source Data file.

proteomics dataset for a single peptide in seconds (Supplementary Fig. 1). In conjunction with cloud-based data storage technology, this new method makes it possible to conveniently query more than one billion indexed MS/MS spectra in PepQueryDB from any computer with internet connection. This radically increased the scalability and usability compared with the previous version of PepQuery. The indexing technology should have wide applications in other computational tools for MS proteomics data analysis. Moreover, it can be equally applied to other MS-based omics technologies, such as metabolomics, to enable fast, targeted data analysis.

We have demonstrated the utility of PepQuery2 in a wide range of applications, but its potential is not limited to these applications. By putting public MS proteomics data directly into the hands of scientists, PepQuery2 will open many new ways to transform these data into useful information for the broad research community.

## Methods

### MS/MS data indexing and the construction of PepQueryDB
Instead of the SQL-based MS/MS indexing method, which was used in the previous version of PepQuery, a completely redesigned indexing scheme was used in PepQuery2. Specifically, for each MS/MS dataset, MS/MS spectra with similar precursor mass after mass rounding and binning are stored in a single compressed MGF file on cloud storage or local storage so that the spectra matched to a query peptide can be retrieved quickly even for large datasets (Supplementary Fig. 1). By default, precursor mass binning of 0.1 Da was used. For example, after mass rounding and binning, MS/MS spectra with precursor mass 1000.11 Da and 1000.12 Da are stored in the file named "10001.mgf.gz" file. The file name after removing the file suffix (.mgf.gz) is the processed mass after mass rounding, and all MS/MS spectra in the same file have the same processed mass. If precursor charge is not available

for a spectrum, charge 2+ and 3+ will be considered by default in the indexing process. With this indexing scheme, for a query peptide sequence, the peptide masses after considering possible fixed and variable modifications were rounded using the same way, then the candidate spectra could be retrieved quickly using the processed peptide mass (Supplementary Fig. 1).

We reconstructed the PepQueryDB using this new indexing method and expanded the database to include more than one billion MS/MS spectra from more than 40 datasets. The expanded database includes all public large CPTAC proteomics datasets. All indexed MS/MS datasets are stored in a public cloud repository. The indexed datasets can be accessed through both the PepQuery2 command line version and the web server version. For each indexed dataset, the parameters used for MS/MS matching are predefined in PepQuery2 based on the experimental protocol for data generation. This removes the requirement to set MS/MS searching parameters by users and thus makes the data usable to a more general audience, including scientists who are not familiar with proteomics experiments and informatics.

### Searching MS/MS data in public proteomics data repositories
Any MS/MS datasets publicly available at PRIDE, MassIVE, jPOSTrepo, or iProX, or USIs from ProteomeXchange, can be directly used by the command line version of PepQuery2. Specifically, for a given dataset with a dataset identifier from these databases, the data files recognized by a specified pattern (such as ".mgf" or "cell_line.*.mgf") are automatically downloaded and indexed. Then the indexed MS/MS data are used for peptide identification in PepQuery2. MS/MS data files in the format of mgf, mzML, mzXML, and raw MS/MS are supported. If the MS/MS data files are raw MS/MS data, ThermoRawFileParser[36] is used to convert the raw MS/MS data to either mgf or mzML format data for MS/MS indexing.

## Support for the search of known peptides or proteins

The previous version of PepQuery only supports the analysis of novel peptides, and PepQuery2 allows using known peptide or protein sequences as a query sequence. In the mode of searching for a known peptide, the peptide is allowed to be present in the reference protein database used by PepQuery2. After digestion of the reference protein database, the query peptides are removed and all other peptides are used for competitive filtering based on reference sequences and competitive filtering based on unrestricted post-translational modification searching[16]. All other processing steps are the same as the novel peptide search workflow.

In the mode of searching for a known protein, the query protein is removed from the reference database. The new reference database without the query protein is then used by PepQuery2. Thus, the query protein is treated as a novel protein and the same workflow for novel protein analysis is used for the search. Only peptides that are unique to the query protein are considered in the search.

## Support for PSM validation

A new mode was implemented in PepQuery2 to support PSM validation. In this mode, both query peptide sequence and the associated spectrum identifiers are required as input. For each query peptide, only the spectra with the specified identifiers are retrieved and scored in PepQuery2. The input PSMs are classified into seven categories on the basis of PepQuery2 identification results, including C1) exact ref match, which means that the query peptide has an exact match to a sequence in the reference database used by PepQuery2. This is only applicable to novel peptide validation; C2) no candidate spectrum, which means there is no spectrum that can be matched to the query peptide based on the peptide mass and allowed mass error tolerance; C3) low score, which indicates that the PSM score computed by PepQuery2 is lower than the prespecified threshold (12 is the default); C4) better ref match, which means that the spectrum can be matched to a peptide in the reference database with an equal or better score. This category is generated based on the result from the competitive filtering based on reference sequences (step 3 in Fig. 1); C5) insignificant score, which means that the PSM fails to pass the statistical evaluation (step 4 in Fig. 1) based on shuffled peptides; C6) better mod ref match, which means that the spectrum can be matched to a reference peptide with a modification that is not typically considered in spectrum-centric database searching. This category is generated based on the result from the competitive filtering through unrestricted modification searching (step 5 in Fig. 1), as described previously[16]. Briefly, all modifications from Unimod are considered in the search. If a spectrum has a better match to a modified peptide from the reference protein database than to the target peptide, the original identification is classified as C6; and C7) confident, which includes PSM passing all the filtering steps as shown in Fig. 1.

## PSM scoring

PepQuery2 uses the same peptide spectrum match (PSM) scoring and statistical evaluation algorithms as described in the original PepQuery publication[16]. In brief, two PSM scoring algorithms, Hyperscore[37] and MVH[38], were implemented. For statistical evaluation of each PSM, randomly shuffled sequences derived from the peptide in the PSM are used to evaluate the statistical significance of the match. The default threshold for p-value filtering is 0.01 for peptides longer than 8 and 0.05 for peptides with length equal or shorter than 8.

## Identifying novel peptides resulted from the *KRAS* G12D mutation

The KRAS G12D mutation peptide LVVVGADGVGK was searched against the global proteome datasets from 10 CPTAC cancer cohorts (CPTAC-BRCA, CPTAC-CCRCC, CPTAC-COAD, CPTAC-GBM, CPTAC-HNSCC, CPTAC-LSCC, CPTAC-LUAD, CPTAC-OV, CPTAC-PDAC and CPTAC-

UCEC). A total of 210,282,541 MS/MS spectra were included in the datasets. The protein database from the GENCODE Human release 34 was used as the reference database. The predefined parameters for each dataset in PepQuery2 were used in the analysis. The *KRAS* mutations detected at the DNA level for tumor samples analyzed in these datasets were downloaded from LinkedOmics [http://linkedomics.org]. Because one TMT sample included multiple tumor samples, a PSM was considered to be supported by genomics data if the *KRAS* G12D mutation was detected at the DNA level in any of the tumor samples included in the TMT experiment from which the PSM was identified.

## Validating novel peptides with W to F substitution

A total of 473 novel peptides with amino acid W to F substitution were collected from a recent publication[21]. These novel peptides were identified in the CPTAC LSCC global proteome dataset[22], in which 3011 PSMs supporting these novel peptides were reported[21]. The PSM level data was obtained through personal communication with the authors of the original study[21]. These PSMs were validated using PepQuery2. Based on the original study[21], the following parameters were used in PepQuery2 validation: Fixed modifications, Carbamidomethyl (C) and TMT 10-plex (K); Variable modifications, Oxidation (M), TMT 10-plex (N-term), TMT 10-plex (S) and Acetylation of peptide N-term; Precursor ion mass tolerance, 20 ppm; MS/MS mass tolerance, 0.05 Da; Enzyme specificity, trypsin; maximum missed cleavages, 2; The range of allowed isotope peak errors, −1,0,1,2,3. The protein database from GENCODE Human release 34 was used as the reference database.

To further assess the quality of peptide identification, the metric derived from peptide retention time prediction based on AutoRT[18] was used. As described in our previous publication[25], an RT prediction model was trained for each run of MS experiment based on the identified peptides from known reference proteins. The training data was generated based on the PSM level identification result from the CPTAC LSCC global proteome dataset in the original study[21]. Only the variable modifications Oxidation (M) and TMT 10-plex (S) were considered. The modifications Carbamidomethyl (C), TMT 10-plex (K) and TMT 10-plex (N-term) were considered as fixed modification in AutoRT model training. Any peptides without the modification TMT 10-plex (N-term) were discarded in RT model training and prediction. For a peptide form with multiple spectra identified, the average RT of these spectra was used as the RT for the peptide form. In model training and testing, any peptides for which the difference between the maximum observed RT and minimum observed RT was > 3 min were removed. In model training, the base size was set to 64 and a maximum of 40 epochs was used. Early stop was also used in model training. After RT models were trained, for each PSM from a novel peptide, the RT of the novel peptide was predicted based on the RT model trained using the data from the run of MS experiment in which the spectrum was identified. Then the absolute difference between the predicted RT and observed RT, i.e., delta RT, was calculated and used as a metric to assess the quality of peptide identification.

A reanalysis was performed by searching the CPTAC LSCC global proteome dataset against a new customized database which contained the customized database used in the original study as well as human reference proteins downloaded from UniProt (downloaded on 12/19/2022, 103,830 proteins). The searching was done through FragPipe (v18.0) powered by the MSFragger[15] (v3.4) search engine and the Philosopher toolkit[39] (v4.4). The parameters were set based on descriptions in the original study[21]. PepQuery validation was performed with the same parameters as described above.

## Known protein validation in small size of MS/MS data

The data for bait protein HDAC1 in AP-MS experiments performed on the 293 T cell line in the BioPlex 3.0[24] project was accessed from MassIVE through the accession number MSV000088555. A total of 86 prey proteins identified from the experiments were downloaded

from https://bioplex.hms.harvard.edu. To validate these prey proteins using PepQuery2, for each protein, all protein isoforms from the UniProt human reference proteome database (04/26/2022) were validated together using PepQuery2. After in silico enzyme digestion, only the peptides uniquely mapped to these protein isoforms were included in the validation, and any peptides that could be mapped to any other proteins were discarded. A peptide shared by different isoforms from the same pray protein but not shared by any other proteins were included in the validation. The following parameters were used in PepQuery2 validation: Variable modifications, Oxidation of M; Precursor ion mass tolerance, 50 ppm; MS/MS mass tolerance, 0.05 Da; Enzyme specificity, trypsin; maximum missed cleavages, 2; The range of allowed isotope peak errors, 0,1. The UniProt human reference proteome database (04/26/2022) was used as the reference database.

### Known protein validation in large size of MS/MS data
Protein identification and quantification results for PGK1, PGK2, NAA10, and NAA11 in the CPTAC LUAD global proteome dataset based on both the original publication[25] and a reanalysis using the FragPipe pipeline [https://fragpipe.nesvilab.org/] were downloaded from LinkedOmics. mRNA quantification for these genes in the same samples were also downloaded from LinkedOmics. The PSMs identifying PGK2 and NAA11 were retrieved and validated by PepQuery2 using the following parameters: Fixed modifications, Carbamidomethyl (C) and TMT 10-plex (K); variable modifications, Oxidation (M), TMT 10-plex (N-term), TMT 10-plex (S); Precursor ion mass tolerance, 20 ppm; MS/MS mass tolerance, 0.05 Da; Enzyme specificity, trypsin; maximum missed cleavages, 1; The range of allowed isotope peak errors, −1,0,1,2,3. The protein database from GENCODE Human release 34 was used as the reference database.

The MS/MS data from the proteome of metastatic cells in colorectal cancer[27] was accessed from MassIVE through the accession number MSV000088431. The reanalysis result of this dataset was downloaded from MassIVE through the accession number RMSV000000617. The PSMs identified from PGK2 were retrieved from this reanalysis and validated using PepQuery2 using the following parameters: fixed modifications, Carbamidomethyl (C); variable modifications, Lysine 13 C(6), Arginine 13 C(6), Oxidation of M, Deamidation of N; Precursor ion mass tolerance, 50 ppm; MS/MS mass tolerance, 0.6 Da; Enzyme specificity, trypsin; maximum missed cleavages, 2. The protein database from GENCODE Human release 34 was used as the reference database.

### Identification of nuORFs in healthy tissue data
A total of 576 nuORFs identified in cancer immunopeptidomics data were collected from a previous study[28]. The protein sequences of these nuORFs were downloaded from MassIVE through the accession number MSV000084787. The raw MS/MS data from a previously published proteomic study analyzing 31 healthy human tissues[29] were downloaded from PRIDE through the accession number PXD010154. Downloaded MS/MS data were indexed and included in the PepQueryDB. The nuORFs were searched against this MS/MS dataset. The predefined parameter set was used. The reference protein database from the previous study was used as the reference database.

### Missing protein identification
The missing proteins were downloaded from neXtProt [https://www.nextprot.org/, 04/14/2022]. The missing proteins classified as PE2 (evidence only at transcript level), PE3 (inferred from homology) or PE4 (proteins inferred to exist) were searched against all MS/MS datasets in PepQueryDB. The predefined parameter set for each dataset was used. In PepQuery2 analysis, all amino acid substitutions from the UniMod database were considered. This was used to filter out any spectra matched to a missing protein but could also be matched equally or better to any other non-missing proteins with an amino acid substitution. Any unique peptides passing the validation of both scoring algorithms (Hyperscore and MVH) implemented in PepQuery2 were included in downstream analyses. A missing protein was considered to be confidently identified if: (1) at least two unique non-nested peptides with length equal or greater than 9 amino acids were identified in the same dataset; and (2) the missing protein was identified in at least two datasets with at least one unique peptide with length equal or greater than 7 amino acids.

### Protein isoform identification
The two protein isoforms of gene *XBP1* (XBP1u and XBP1s) were downloaded from UniProt. The isoform XBP1u (P17861-1) is also known as unprocessed XBP1, whereas the isoform XBP1s (P17861-2) is also known as processed XBP1. The two isoforms were searched against all MS/MS datasets in PepQueryDB. The predefined parameter set for each dataset was used. The SwissProt human protein database, including protein isoforms (05/17/2022) was used as the reference database.

### Reporting summary
Further information on research design is available in the Nature Portfolio Reporting Summary linked to this article.

## Data availability
The 48 MS/MS datasets indexed in PepQueryDB were downloaded from PDC, PRIDE, or MassIVE. Detailed dataset information, including hyperlinks to access the datasets, is available in Supplementary Data 1. The *KRAS* mutations detected at the DNA level for CPTAC tumor samples were downloaded from LinkedOmics. The human reference proteins used in the reanalysis of the CPTAC LSCC global proteome dataset were downloaded from UniProt on 12/19/2022. The AP-MS data from BioPlex 3.0[24] was downloaded from MassIVE through the accession number MSV000088555. The human reference proteins used in the analysis of the AP-MS data were downloaded from UniProt on 04/26/2022. The MS/MS data from the proteome of metastatic cells in colorectal cancer[27] was accessed from MassIVE through the accession number MSV000088431, and the protein database from GENCODE Human release 34 was used as the reference database for this analysis. The protein sequences of the nuORFs were downloaded from MassIVE through the accession number MSV000084787. The raw MS/MS data from the analyzing 31 healthy human tissues were downloaded from PRIDE through the accession number PXD010154. The missing proteins were downloaded from neXtProt (04/14/2022). For the XBP1 analysis, the SwissProt human protein database, including protein isoforms (05/17/2022) was used as the reference database. Source data are provided with this paper.

## Code availability
Both the command line version and the web version of PepQuery2 are available at [http://www.pepquery.org]. The source code of PepQuery2 is available at [https://github.com/bzhanglab/PepQuery]. Scripts used in the manuscript are available at https://github.com/wenbostar/pepquery2_manuscript.

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

## Acknowledgements

This study was supported by the National Cancer Institute (NCI) CPTAC award U24 CA210954 (B.Z.), the Cancer Prevention & Research Institutes of Texas (CPRIT) award RR160027 (B.Z.), and funding from the McNair Medical Institute at The Robert and Janice McNair Foundation (B.Z.). We gratefully acknowledge the support of NVIDIA Corporation with the donation of the Titan Xp GPU used for this research. BZ is a CPRIT Scholar in Cancer Research and a McNair scholar.

## Author contributions

B.W. and B.Z. conceived the study. B.W. developed the software and performed all data analysis. B.W. and B.Z. interpreted the data and wrote the manuscript.

## Competing interests

B.Z. received consulting fee from AstraZeneca. The remaining author declares no competing interests.
