## [Peer Review File · Nature Communications]

PepQuery2 democratizes public MS proteomics data for rapid peptide searchingREVIEWER COMMENTS

Reviewer #1 (Remarks to the Author):

In this manuscript, "PepQuery2 democratizes public MS proteomics data for rapid peptide searching," the authors present a computational method to query for the evidence of a peptide from tandem mass spectra at a repository scale. At the core of PepQuery2 is an indexing strategy whereby tandem mass spectra are indexed by their precursor mass, such that relatively few tandem mass spectra must be retrieved for any given query peptide. I found the manuscript to be interesting and well-written; however, I think it needs more detailed methods and I am left with a few questions and issues to address. I would recommend it be accepted with major revisions.

1. The "C1: exact ref match" classification seems better suited as an error or warning for the user, rather than a category of its own. Is there a case where a C1 classification is useful by itself?

2. How are the p-values for a peptide query calculated? Have the p-values been confirmed to be valid? How are decoys generated for this procedure?

3. Given the statistical validation provided by the p-value in the "C5: insignificant score" category, it seems that the "C3: low score" category is unnecessary; any query of sufficiently low score should have a poor p-value. Is there a reason to keep C3 as a separate category?

4. How is the "mod ref match" step performed?

5. I found the the "Validating novel peptide identifications" section to be unconvincing. The original spectrum-centric database search was performed incorrectly in the original study, as noted by the authors, "...these potential false identifications from the original study can be attributed to the lack of competition from peptides containing amino acid W because in the customized database used in the original study, all Ws were replaced with an F without keeping a version of the unaltered sequence." It would be much more compelling to compare the PepQuery2 results against a spectrum-centric search that correctly considered both Ws and Fs.

6. The manuscript claims that peptide-centric searching is an orthogonal method to spectrum-centric searching. I would disagree, because both rely on evidence from the same data rather than an alternative line of evidence. Furthermore, performing a peptide-centric analysis on the same data that we've performed a spectrum-centric analysis is a case of statistical "double-dipping": querying a peptide that was already considered in a the spectrum-centric search may merely ignore the other hypotheses tested in the spectrum-centric search.

7. Were isotope errors considered in the original BioPlex analysis? In figure 4, I suspect that the deamidated CHD4 peptide may actually be the result of isolating the C13 peak of the unmodified peptide.

8. The manuscript states that "PepQuery2 does not rely on the target-decoy strategy" in the "Validating known peptide identifications" section, yet in the methods it describes using a decoy approach for statistical validation: "C5) insignificant score, which means that the PSM fails to pass the statistical evaluation based on shuffled peptides ($p < 0.01$ is the default)".

9. For the NAA10 vs NAA11 example presented in Supplementary Figure 6, why did the NAA11 peptide score higher than the NAA10 peptide in the original database search?

Reviewer #2 (Remarks to the Author):

The authors present the second version of PepQuery. This tool enables researchers to search a large quantity of (mostly) publicly available with a peptide query. The local version includes all the major public repositories. While the web version allows for easy querying of a number of data sets from these public repositories. The ability for researchers to search for a peptide in such large repositories has enormous potential for a number of applications, e.g., target validation.

The manuscript is well written and the tools actually work. The latter should not sound like a surprise, but academic software is notorious for not (easily) working. Although the code on github is mostly undocumented. Since documentation is not standard for academic software, this is not a concern, but if the time is available for the authors I would recommend to provide more documentation in the code.

I think the tool is a great resource for the community, I suggest a major revision. For publication I would like the authors to respond to my concerns outlined below.

Major concerns

1. It is not clear to me why PepQuery2 suffers less from false hits than spectrum-centric approaches. If the W->F codon reassignment (and "native" form) is part of the search space of the spectrum-centric tools it should favor the highest scoring PSM for that spectrum. This problem sounds like an issue with the scoring function of the search engine rather than PepQuery2 solving this problem. I suspect PepQuery2 is more conservative in terms of assigning a hit. For a fair comparison in this case the FDR threshold for the spectrum-centric approach should be set more conservative too.

2. How was PeptideProphet able to score the PSM higher when using PepQuery2? Either this option was previously not shown to PeptideProphet (i.e., scoring function problems) or there is additional scoring from PepQuery2 side that is not available for PepQuery2. In the latter case, please explain what is provided additionally to score the PSM.

3. If I understand correctly the search was incorrectly performed without retaining the unaltered version. I would argue this search was performed wrong (even though not part of this article, if I understand correctly). A search should always be performed with all expected peptidofoms or false hits can be expected. I suggest the authors redo the search with the correct database and report how for example wrong database use can be avoided with PepQuery2.

4. Usually when performing peptide validation not only the first hit of a spectrum is considered, if both "AVVLMSHLGRPDGVPMPDK" and "SVVLMSHLGRPDGVPMPDK" are close in terms of score this should be picked up. It is not clear to me how PepQuery2 is better in identifying cases where very similar peptides have near identical scores (high ambiguity). As far as I understand both methods still require a manual inspection for assignment of the correct PSM. If this is based on majority voting by earlier observations, are there biases introduced in the identifications. How can this influence down-stream analysis?

5. PepQuery2 does not consider all potential PSMs, traditional does consider all PSMs as long as they are in the search space (and thus in the database). How does PepQuery2 effectively control the FDR when not having access to other potentially high-scoring PSMs? In this case I am especially referring to false negatives, or if the thresholding is less conservative also for false positives.

6. Please elaborate more on how the hyperscore was implemented (equation and threshold).

Minor concern

1. Figure 4, it is probably to not use the non-identified data points into account when calculating the spearman correlation.

Reviewer #3 (Remarks to the Author):

In this manuscript, Wen and Zhang present an updated version of PepQuery that allows the rapid query of millions of data sets with a peptide sequence. The approach of PepQuery, allowing the dynamic query of public and private proteomic data, without having to rerun a database search, is rather unique and of great value to the community. I have therefore no doubt concerning the scientific value of this research. But the paper and implementation do not allow the reader to fully benefit from this work.

Before the manuscript can be accepted, I recommend the consideration of the following points:

Major concerns

1- I tried the tool mainly from the web interface, and I can confirm that the query is fast and intuitive. But for most of my queries (90%), the interface became unresponsive or threw an error that I could not resolve: "An error has occurred. Check your logs or contact the app author for clarification.". I acknowledge the difficulty of running a web service as part of a research group. But the web interface is such an essential component of this paper, it ought to work more reliably. In my case it systematically stopped working after the first query, the error message did not allow me to troubleshoot the problem.

2- When the tool did not hang forever, I searched sequences identified from a database search. I could find the most confident hits, and appreciate the quality of the spectrum annotation. But for the less confident ones that I was especially interested in, the tool simply lists "No result!". It is not clear to me whether no result is returned because the tool is not working, or whether because the peptide is excluded by some internal threshold that is not communicated to the user. This needs to be fixed.

3- The paper must describe the implementation in sufficient detail so that the reader can understand what is happening without having to read the other PepQuery papers. In Particular, the authors need to detail how spectra are annotated, how scores are computed, and what thresholds are used. The authors also need to describe the output of the tool and the content of the web interface.

4- A central component of the study is the validation of the peptides and PSMs. The authors mention implementing a novel validation approach, partitioning the PSMs into seven categories, one of which including "PSM passing PepQuery2 validation." but no information is provided on what the validation refers to. The authors need to detail how validation is done, and provide some information to the reader on how to interpret these categories.

5- The manuscript makes a strong point of examples where they were able to identify peptides that were a "better match" than the ones reported in publications. But the match being better is highly subjective of the score and spectrum annotation used. The examples presented in the manuscript are very clear. But often the difference between two matches is subtle - when does it become significant? It is not clear how the authors draw the line to consider that a match is better, this is extremely important to clarify.

6- The authors have implemented different scores, and when two peptides present similar scores, they refer to the precursor mass deviation. How are these different features taken into account in the validation? Implementing something like Percolator would strengthen the approach.

Minor points

1- Please briefly explain c1-7 categories and how to interpret them before discussing them.

2- Please rephrase "is hard to distinguish from the KRASG12D peptide in mass spectrum."

Re: NCOMMS-22-29704 “PepQuery2 democratizes public MS proteomics data for rapid peptide searching”

REVISIONS IN RESPONSE TO REVIEWERS’ COMMENTS

We thank the reviewers for the insightful comments and constructive suggestions. We have considered all comments and suggestions and revised the manuscript accordingly. For your convenience, we have also included a version with “tracked changes” in the submission. Please see below for a point-by-point response to each of the points made by the reviewers.

Reviewer #1 (Remarks to the Author):

In this manuscript, "PepQuery2 democratizes public MS proteomics data for rapid peptide searching," the authors present a computational method to query for the evidence of a peptide from tandem mass spectra at a repository scale. At the core of PepQuery2 is an indexing strategy whereby tandem mass spectra are indexed by their precursor mass, such that relatively few tandem mass spectra must be retrieved for any given query peptide. I found the manuscript to be interesting and well-written; however, I think it needs more detailed methods and I am left with a few questions and issues to address. I would recommend it be accepted with major revisions.

Response: We thank the reviewer for the positive comments and insightful questions.

1. The "C1: exact ref match" classification seems better suited as an error or warning for the user, rather than a category of its own. Is there a case where a C1 classification is useful by itself?

Response: The C1 group is essentially invalid identifications for novel peptides, however, these errors cannot be identified without checking. The first step in PepQuery2 is to simply check the candidate novel peptides against a selected reference protein database, and any exact matches are classified in the C1 category. This often happens when the database used in the original study is less inclusive than the one selected in PepQuery2 (e.g., UniProt vs Ensembl). Moreover, some mistakes could be introduced in reference database preparation, such as the study identifying novel peptides resulted from tryptophan-to-phenylalanine codon reassignment. We agree with the reviewer that these are just “errors”, but the fact that the W2F results were published in *Nature* after peer review suggests that it is not easy to eyeball these problems, and a formal check is necessary to identify them. Keeping them as a separate class will allow users to quickly identify these errors.

We clarified the definition of C1 and other groups in the revised manuscript:

Page 5: “C1 (exact ref match) includes input PSMs for which the peptide has an exact match to a sequence in the reference database. This is only applicable to novel peptide validation, and PSMs in this category are essentially invalid novel identifications. This could happen when the input PSMs are identified using a different database than the selected reference database in PepQuery2 analysis.”

2. How are the p-values for a peptide query calculated? Have the p-values been confirmed to be valid? How are decoys generated for this procedure?

Response: The details about p-value calculation and validation were described in the original PepQuery paper published in *Genome Research* (Wen B, Wang X, Zhang B. *Genome research*, 2019, 29(3): 485-493). The p-value calculation in PepQuery2 uses the same method. For each PSM passing step 3 filtering as shown in Figure 1, randomly shuffled sequences derived from the peptide in the PSM are used to evaluate the statistical significance of the match. Specifically, a specified number of unique random peptides (e.g., 10,000) are generated by randomly shuffling of the original peptide sequence. The resulted random peptide sequences have the same amino acid composition as the original sequence. For each random peptide, the Hyperscore or the MVH score is calculated to quantify the match between the random peptide and the spectrum in the PSM. Based on each of the scoring algorithms, a p-value is then calculated for the PSM:

$$\text{p-value} = \frac{N_s + 1}{N},$$

where N_s is the number of random peptides with a higher score than the original PSM scores, and N is the total number of random peptides generated. Only PSMs with a p-value ≤ 0.01 are retained for the unrestricted modification searching-based filtering. Since PepQuery uses a series of filtering steps including the p-value and other steps as shown in Figure 1, accuracy of the combined filtering was systematically validated using different methods in the original paper.

In the revised manuscript, to allow readers to understand this without going back to the original paper, we have added a new subsection to Methods to briefly describe both PSM scoring and p-value calculation algorithms:

Page 15: “*PSM scoring PepQuery2 uses the same peptide spectrum match (PSM) scoring and statistical evaluation algorithms as described in the original PepQuery publication¹⁶. In brief, two PSM scoring algorithms, Hyperscore³⁶ and MVH³⁷, were implemented. For statistical evaluation of each PSM, randomly shuffled sequences derived from the peptide in the PSM are used to evaluate the statistical significance of the match. The default threshold for p-value filtering is 0.01.*”

3. Given the statistical validation provided by the p-value in the "C5: insignificant score" category, it seems that the "C3: low score" category is unnecessary; any query of sufficiently low score should have a poor p-value. Is there a reason to keep C3 as a separate category?

Response: It is true that a query with sufficiently low score should have a poor p-value. The reason to keep C3 is to speed up the analysis. Specifically, if the score for a query is lower than a specified low score threshold, spectrum associated with this query will be categorized as C3 and will not be subjected to further analyses since the match is of low quality. This was clarified in the revised manuscript.

Page 5: “*C3 (low score) includes input PSMs with a PepQuery2 computed PSM score lower than the prespecified threshold (Methods). These low-quality matches are excluded from further analysis to save time.*”

4. How is the "mod ref match" step performed?

Response: The “mod ref match” step is a competitive filtering step based on unrestricted modification searching. The detail about this step was described in the original PepQuery paper as the following: “*All spectra involved in the remaining PSMs are searched against the selected protein reference database while considering all modifications from the Unimod database (<http://www.unimod.org/>) except for amino acid substitutions. Using the same scoring algorithm, if a spectrum has a better match to a modified peptide from the reference protein database than to the target peptide, the original identification is rejected. To speed up the searching, a peptide index and a modification index are generated. For a given protein reference database and user-specified fixed modifications and digestion parameters, a peptide index is generated for nonredundant peptides as shown in Figure 2B. This index is a hash map in which the integer values of the peptide masses are the key and the corresponding peptide sequences and masses are the values. The peptide indexing takes just a few seconds on a typical computer. The modification index is a hash map in which integer values of the modification masses are the key and the corresponding modification objects are the values as shown in Figure 2C.*”.

In the revised manuscript, we have added a brief description:

Page 14-15: “*C6) better mod ref match, which means that the spectrum can be matched to a reference peptide with a modification that is not typically considered in spectrum-centric database searching. This category is generated based on the result from the competitive filtering through unrestricted modification searching (step 5 in Figure 1), as described previously¹⁶. Briefly, all modifications from Unimod are considered in the search. If a spectrum has a better match to a modified peptide from the reference protein database than to the target peptide, the original identification is classified as C6;*”

5. I found the the "Validating novel peptide identifications" section to be unconvincing. The original spectrum-centric database search was performed incorrectly in the original study, as noted by the authors, "...these potential false identifications from the original study can be attributed to the lack of competition from peptides containing amino acid W because in the customized database used in the original study, all Ws were replaced with an F without keeping a version of the unaltered sequence." It would be much more compelling to compare the PepQuery2 results against a spectrum-centric search that correctly considered both Ws and Fs.

Response: One of the important applications of PepQuery2 is to validate novel peptides identified using spectrum-centric search methods. To identify novel peptides, the traditional spectrum-centric search methods require the construction of a customized protein database, and errors could be introduced in this step, as shown in this study identifying novel W2F peptides. Novel peptide validation using PepQuery2 does not require customized protein database construction and thus reduce the chance of errors. This example is used in part to illustrate this unique advantage of PepQuery2.

We agree with the reviewer that it would be interesting to compare the PepQuery2 results against a spectrum-centric search that correctly considered both Ws and Fs. This has been added to the revised manuscript:

Methods section, Page 16: “A reanalysis was performed by searching the CPTAC LSCC global proteome dataset against a new customized database which contained the customized database used in the original study as well as human reference proteins downloaded from UniProt (downloaded on 12/19/2022, 103,830 proteins). The searching was done through FragPipe (v18.0) powered by the MSFragger¹⁵ (v3.4) search engine and the Philosopher toolkit³⁹ (v4.4). The parameters were set based on descriptions in the original study²¹. PepQuery validation was performed with the same parameters as described above.”

Results section: Page 7: “Repeating the spectrum-centric analysis using a customized database including human reference proteins downloaded from UniProt (**Methods**) reduced the total number of candidate peptides with W>F substitution from 473 to 240 (1024 PSMs) and the proportion of the C4 group in PepQuery2 validation to 27% (**Supplementary Figure 2b**). Despite these improvements, there were still many candidate PSMs failing PepQuery2 validation, including 510 PSMs (50%) for which the spectrum was matched to a reference peptide with a modification not considered in the spectrum-centric analysis (C6). Together, these results demonstrate PepQuery2 as an effective tool for identifying mistakes in customized database construction (a task not required for PepQuery2 analysis) as well as potential false positives among the novel peptides identified in spectrum centric analysis.”

6. The manuscript claims that peptide-centric searching is an orthogonal method to spectrum-centric searching. I would disagree, because both rely on evidence from the same data rather than an alternative line of evidence. Furthermore, performing a peptide-centric analysis on the same data that we've performed a spectrum-centric analysis is a case of statistical "double-dipping": querying a peptide that was already considered in a the spectrum-centric search may merely ignore the other hypotheses tested in the spectrum-centric search.

Response: We agree “orthogonal” may be misleading, and we have removed the word from the revised manuscript.

7. Were isotope errors considered in the original BioPlex analysis? In figure 4, I suspect that the deamidated CHD4 peptide may actually be the result of isolating the C13 peak of the unmodified peptide.

Response: In the original BioPlex analysis, through a personal communication with the author, isotope errors were not considered at database searching level. Instead, a custom algorithm was used to check the monoisotopic peak assignments for the precursor of each MS/MS spectrum and then the precursor mass of each MS/MS spectrum was corrected, if necessary, before performing database searching. The algorithm for correcting monoisotopic mass assignments was described in a later publication (*Rad R, Li J, Mintseris J, et al., Journal of Proteome Research, 2020, 20(1): 591-598.*) from the same lab.

During the revision, we have compared the matches between the deamidated CHD4 peptide and the unmodified version against the same spectrum. As shown below, the match from the

deamidated CHD4 peptide (top part) looks better than that from the unmodified version (bottom part). This suggests that the spectrum is more likely derived from the deamidated form.

8. The manuscript states that "PepQuery2 does not rely on the target-decoy strategy" in the "Validating known peptide identifications" section, yet in the methods it describes using a decoy approach for statistical validation: "C5) insignificant score, which means that the PSM fails to pass the statistical evaluation based on shuffled peptides ($p < 0.01$ is the default)".

Response: Thanks for pointing out this issue. We're sorry for this confusion. The target-decoy strategy we mentioned in that section is the traditional target-decoy strategy used for global FDR estimation. It is different from the shuffled peptide-based statistical evaluation which is used to generate p-values for individual PSMs. To make it clearer, in the revised manuscript, we have reworded the sentence as "PepQuery2 does not rely on the traditional target-decoy strategy typically used for global FDR estimation" (page 8).

9. For the NAA10 vs NAA11 example presented in Supplementary Figure 6, why did the NAA11 peptide score higher than the NAA10 peptide in the original database search?

Response: The reason that FragPipe didn't identify this spectrum as an NAA10 peptide is because that there are two "variable" modifications on the first amino acid (Oxidation and TMT on N-term, in this case FragPipe counts two modifications on M, which is the first amino acid) so this peptide form is not in the search space in the FragPipe database searching (only one variable modification on each amino acid). We confirmed this with the developer of FragPipe during the revision.

PepQuery doesn't treat this as two modifications on M so this form is included in the PepQuery search space. When we did a FragPipe search by setting TMT N-term modification as fixed modification, then FragPipe reported the NAA10 as the best match for this spectrum.

Reviewer #2 (Remarks to the Author):

The authors present the second version of PepQuery. This tool enables researchers to search a large quantity of (mostly) publicly available with a peptide query. The local version includes all the major public repositories. While the web version allows for easy querying of a number of data sets from these public repositories. The ability for researchers to search for a peptide in such large repositories has enormous potential for a number of applications, e.g., target validation.

The manuscript is well written and the tools actually work. The latter should not sound like a surprise, but academic software is notorious for not (easily) working. Although the code on github is mostly undocumented. Since documentation is not standard for academic software, this is not a concern, but if the time is available for the authors I would recommend to provide more documentation in the code.

I think the tool is a great resource for the community, I suggest a major revision. For publication I would like the authors to respond to my concerns outlined below.

Response: We thank the reviewer for the positive comments and for understanding the challenges in developing and maintaining academic software.

Major concerns

1. It is not clear to me why PepQuery2 suffers less from false hits than spectrum-centric approaches. If the W->F codon reassignment (and “native” form) is part of the search space of the spectrum-centric tools it should favor the highest scoring PSM for that spectrum. This problem sounds like an issue with the scoring function of the search engine rather than PepQuery2 solving this problem. I suspect PepQuery2 is more conservative in terms of assigning a hit. For a fair comparison in this case the FDR threshold for the spectrum-centric approach should be set more conservative too.

Response: As described in the original paper of PepQuery (*Wen B, Wang X, Zhang B, Genome research, 2019, 29(3): 485-493*) and briefly outlined in Figure 1 in the current manuscript, a series of filtering steps are implemented to remove potential false hits to a target peptide of interest. These include statistical validation evaluation for individual PSMs and competitive filtering based on unrestricted post-translational modification searching, which are not applicable in typical spectrum-centric analysis. Although some spectrum-centric search engines such as Open-pFind and MSFragger have implemented an open search algorithm that could consider all possible modifications, it is not used in typical proteomics data analysis. We showed the utility of PepQuery in reducing false hits in the original PepQuery publication, and this was further systematically demonstrated in additional datasets using an independent and quantitative metric derived from deep learning-based retention time prediction (*Wen B, Li K, Zhang Y, et al., Nature communications, 2020, 11(1): 1-14.*). These were briefly introduced in the introduction section.

We agree with the reviewer that if the reference database contains the “native form”, as it should do, some of the spectra will likely be scored higher to native forms instead of the W2F peptides (see our response to comment #3). Unfortunately, this error did happen and was not caught by peer review, leading to potentially misleading conclusion in a paper published in a high-profile journal. It is our hope that by removing the customized database construction step in novel peptide validation, PepQuery reduces the chance of errors in customized database preparation and thus facilitates systematic identification of such errors.

In addition, in the original W2F paper, global FDR control (1%) was applied to all identified peptides, including novel peptides. It is well acknowledged in the proteomics community that with global FDR control, the actual FDR for the group of novel peptides (W2F peptides in this case) is expected to be higher, we have previously demonstrated that PepQuery can effectively remove potential false hits in this scenario (*Wen B, Li K, Zhang Y, et al., Nature communications, 2020, 11(1): 1-14.*).

We would like to clarify that the goal of PepQuery2 is not to compete with spectrum-centric search tools. Instead, one of the major utilities of PepQuery is to serve as a complementary tool to validate spectrum-centric search results.

2. How was PeptideProphet able to score the PSM higher when using PepQuery2? Either this option was previously not shown to PeptideProphet (i.e., scoring function problems) or there is additional scoring from PepQuery2 side that is not available for PeptideProphet. In the latter case, please explain what is provided additionally to score the PSM.

Response: We think this is primary due to the difference of scoring algorithms. It has been reported that the scores from different scoring algorithms may not always agree with each other (*Cox J, Neuhauser N, Michalski A, et al., Journal of proteome research, 2011, 10(4): 1794-1805; Alves G, Ogurtsov A Y, Yu Y K, PloS one, 2010, 5(11): e15438*). Therefore, it is possible that the peptide matches rank differently using different tools, especially for matches with relative low scores. The original W2F study used MSFragger as the search engine. Because MSFragger is not open source, we don't know the exact spectrum preprocessing and scoring algorithms. PepQuery implemented two scoring algorithms, Hyperscore and MVH. The equations are included in the original publication and the actual implementation can be checked in our source code in <https://github.com/bzhanglab/PepQuery> (also see our response to comment #6).

3. If I understand correctly the search was incorrectly performed without retaining the unaltered version. I would argue this search was performed wrong (even though not part of this article, if I understand correctly). A search should always be performed with all expected peptidofoms or false hits can be expected. I suggest the authors redo the search with the correct database and report how for example wrong database use can be avoided with PepQuery2.

Response: We completely agree with the reviewer on the importance of using the right database for analysis. Following this suggestion, we reanalyzed the data using the correct database and applied PepQuery2 to the analysis results. This has been added to the revised manuscript:

Methods section, Page 16: “A reanalysis was performed by searching the CPTAC LSCC global proteome dataset against a new customized database which contained the customized database used in the original study as well as human reference proteins downloaded from UniProt (downloaded on 12/19/2022, 103,830 proteins). The searching was done through FragPipe (v18.0) powered by the MSFragger¹⁵ (v3.4) search engine and the Philosopher toolkit³⁹ (v4.4). The parameters were set based on descriptions in the original study²¹. PepQuery validation was performed with the same parameters as described above.”

Results section: Page 7: “Repeating the spectrum-centric analysis using a customized database including human reference proteins downloaded from UniProt (**Methods**) reduced the total number of candidate peptides with $W>F$ substitution from 473 to 240 (1024 PSMs) and the proportion of the C4 group in PepQuery2 validation to 27% (**Supplementary Figure 2b**). Despite these improvements, there were still many candidate PSMs failing PepQuery2 validation, including 510 PSMs (50%) for which the spectrum was matched to a reference peptide with a modification not considered in the spectrum-centric analysis (C6). Together, these results demonstrate PepQuery2 as an effective tool for identifying mistakes in customized database construction (a task not required for PepQuery2 analysis) as well as potential false positives among the novel peptides identified in spectrum centric analysis.”

4. Usually when performing peptide validation not only the first hit of a spectrum is considered, if both “AVVLMShLGRPDGVPMPDK” and “SVVLMShLGRPDGVPMPDK” are close in terms of score this should be picked up. It is not clear to me how PepQuery2 is better in identifying cases where very similar peptides have near identical scores (high ambiguity). As far as I understand both methods still require a manual inspection for assignment of the correct PSM. If this is based on majority voting by earlier observations, are there biases introduced in the identifications. How can this influence down-stream analysis?

Response: In PepQuery2, for a spectrum matched to a target peptide, such as a novel peptide, if there is a different peptide from the reference database matched to the spectrum with an equal or higher score than the match to the target peptide, the spectrum matched to the target peptide is considered as an unconfident match even if the two peptides have equal scores to the spectrum. We prefer to use this conservative criterium in the validation of novel or unexpected peptide identifications because the prior probability of observing novel or unexpected peptides is much lower than the competing reference peptides. These identifications should at least require more careful investigation.

In contrary, when the match to the target peptide is better than any peptides in the reference database and p-value is less than 0.01, this is considered as a putative confident match even if the score of the target peptide match is close to the best match from the reference database. The putative match is further evaluated using unrestricted modification searching, in which about 1000 modifications are considered and many peptide forms from reference database with different modifications are competed with the target peptide. This further enhances the reliability of the target peptide identification, and we have previously demonstrated its utility in reducing false positives in variant peptide identification (*Wen B, Li K, Zhang Y, et al., Nature communications, 2020, 11(1): 1-14.*).

We agree with the reviewer that when two peptides have close scores to the same spectrum, it is very useful to manually check the matches. In PepQuery, we export all details required for manual checking. The web server of PepQuery2 (<http://pepquery2.pepquery.org/>) provides annotated spectra for best matches from the competitive filtering based on reference sequences and competitive filtering based on unrestricted post-translational modification searching for visual check of these matches. Details can be found at <http://pepquery.org/document.html#webresult>. For the standalone version of PepQuery2, the result can be imported PDV (<https://github.com/wenbostar/PDV>) for visualization. With PDV, each spectrum matched to a target peptide can be visualized by comparing to any reference peptides matched by PepQuery2. Detailed information can be found at <http://pepquery.org/document.html#savis>. These have been clarified in Discussion in the revised manuscript.

Page 11-12: *“Both PepQuery and PepQuery2 use stringent criteria for novel peptide validation. When a candidate novel peptide and a reference peptide have equal matching score to a spectrum, the spectrum is preferentially associated with the reference peptide based on the consideration that the prior probability of observing a novel peptide is much lower than a reference peptide. The competitive filtering step based on unrestricted modification searching further excludes the possibility that the spectrum has a better match to a reference peptide sequence with a modification not considered in the spectrum-centric analysis. When two peptides have equal or close scores to the same spectrum, it is useful to manually check the matches. The web server of PepQuery2 provides annotated spectra for visual check of the matches. For the standalone version, details required for manual checking can be exported for visualization using PDV³⁵. In this study, we also identified independent evidence to support PepQuery2 results. For the W2F peptide validation, we used a metric derived from retention time prediction to evaluate the quality of the results. For the cases of PGK2 and NAA11, we made use of paired mRNA data to support PepQuery2 results. For CHD5, we used prior knowledge about protein complex. Such analyses are very helpful in evaluating matches with equal or similar scores.”*

5. PepQuery2 does not consider all potential PSMs, traditional does consider all PSMs as long as they are in the search space (and thus in the database). How does PepQuery2 effectively control the FDR when not having access to other potentially high-scoring PSMs? In this case I am especially referring to false negatives, or if the thresholding is less conservative also for false positives.

Response: PepQuery is a targeted peptide search engine. For each target peptide, PepQuery considers all spectra within a precursor mass tolerance window of the target peptide, which can be determined based on the instrument setting for generating the data. Spectra outside the precursor mass tolerance window are unlikely to be true matches, so possibility of false negative based on the search space is minimal.

Due to the nature of targeted analysis, the traditional target-decoy strategy used for global FDR control is not applicable. For statistical evaluation of each candidate PSM, randomly shuffled sequences derived from the peptide in the PSM are used to evaluate the statistical significance of the match. Specifically, a specified number of unique random peptides (e.g., 10,000) are generated by randomly shuffling of the original peptide sequence. The resulted random peptide sequences have the same amino acid composition as the original sequence. For each random peptide, the

Hyperscore or the MVH score is calculated to quantify the match between the random peptide and the spectrum in the PSM. Based on each of the scoring algorithms, a p-value is then calculated for the PSM:

$$p\text{-value} = \frac{N_s+1}{N},$$

where N_s is the number of random peptides with a higher score than the original PSM scores, and N is the total number of random peptides generated. Only PSMs with a p-value ≤ 0.01 are retained for further analysis. In addition to this statistical evaluation, PepQuery also includes a series of filtering steps such as the unrestricted modification searching-based filtering, which further improves the reliability of the novel findings.

This process is described in detail in the original PepQuery publication and briefly illustrated in Figure 1 of this manuscript. Only PSMs passing all the filtering steps as shown in Figure 1 are considered as confident identifications. Sensitivity and specificity of the method were assessed using multiple methods in the original PepQuery publication and a follow-up paper (*Wen B, Li K, Zhang Y, et al., Nature communications, 2020, 11(1): 1-14.*)

6. Please elaborate more on how the hyperscore was implemented (equation and threshold).

Response: As mentioned in our response to comment #2, the implementation of hyperscore and MVH score was described in detail in the method section of the original paper of PepQuery. In the revised manuscript, we have added a sentence to the method section to clarify that the detail of hyperscore implementation was described previously with a citation on Page 15. For the convenience of the reviewer, we also copied the text from the original publication below:

The Hyperscore calculation is similar to X!Tandem (Craig and Beavis 2004):

$$\text{hyperscore} = \log \left(N_b! N_y! \sum_{i=1}^{N_b} I_{b,i} \sum_{i=1}^{N_y} I_{y,i} \right)$$

where N_b is the number of matched b-ions, N_y is the number of matched y-ions, $I_{b,i}$ are the intensities of matched b-ions, and $I_{y,i}$ are the intensities of matched y-ions.

We don't have a threshold of hyperscore to define a high confident match. Instead, we derive a p-value for each peptide spectrum match to evaluate the statistical significance of the match. Then use the p-value filtering (e.g., p-value ≤ 0.01) along with other competitive filtering steps as illustrated in Figure 1 to define a confident match.

Minor concern

5. Figure 4, it is probably to not use the non-identified data points into account when calculating the spearman correlation.

Response: For protein abundance correlation shown in Figure 4d, only samples with non-missing values in both samples were considered when calculating the spearman correlation. For mRNA abundance correlation shown in the revised Figure 4e, we also excluded samples with 0 values in

either of the two genes when calculating the spearman correlation and clarified this in the figure legend.

Reviewer #3 (Remarks to the Author):

In this manuscript, Wen and Zhang present an updated version of PepQuery that allows the rapid query of millions of data sets with a peptide sequence. The approach of PepQuery, allowing the dynamic query of public and private proteomic data, without having to rerun a database search, is rather unique and of great value to the community. I have therefore no doubt concerning the scientific value of this research. But the paper and implementation do not allow the reader to fully benefit from this work.

Response: We thank the reviewer for the positive comments and very helpful suggestions.

Before the manuscript can be accepted, I recommend the consideration of the following points:

Major concerns

- 5- I tried the tool mainly from the web interface, and I can confirm that the query is fast and intuitive. But for most of my queries (90%), the interface became unresponsive or threw an error that I could not resolve: “An error has occurred. Check your logs or contact the app author for clarification.” I acknowledge the difficulty of running a web service as part of a research group. But the web interface is such an essential component of this paper, it ought to work more reliably. In my case it systematically stopped working after the first query, the error message did not allow me to troubleshoot the problem.

Response: We’re sorry for this issue. First, we would like to clarify that the web version is implemented using Shiny and Shiny Server and it is hosted on a single server. To speed up the search for each job, only one job can be submitted to run and it will use all available CPUs in our server. When a job is running, the web server cannot be accessed by other users. We have included this information in the documentation (<http://pepquery.org/document.html>, Web application section). We are considering adding more nodes for the web server in the future to allow running multiple jobs at the same time.

During the revision, we have implemented a more informative error reporting system. Our lab members and outside collaborators have performed many tests. Based on the feedback, we have fixed a few bugs and made some modifications to improve the web server (see News section at <http://pepquery.org/index.html>). We believe the web version should be much more robust now.

We understand it is impossible to identify and fix all bugs. To address this problem, we do have the Github issue system (<https://github.com/bzhanglab/PepQuery/issues>), which allows users to report issues they encountered during the use of the web server or the standalone version.

- 2- When the tool did not hang forever, I searched sequences identified from a database search. I could find the most confident hits, and appreciate the quality of the spectrum annotation. But for

the less confident ones that I was especially interested in, the tool simply lists “No result!”. It is not clear to me whether no result is returned because the tool is not working, or whether because the peptide is excluded by some internal threshold that is not communicated to the user. This needs to be fixed.

Response: In the web server, to simply the result visualization, only PSMs passing the filtering in step 3 are shown. During the revision, we have made a change to the web server to show all matches when there is any match passing the step 3 filtering. If there is no match passing step 3 filtering, detailed information from the PepQuery command line output will be shown (see screenshot below). From this, users will be able to know why there is no result returned.

```
Identification result

There is no any match found. Detailed log about the search can be found below:

- Load 12 parameter sets.
- Load 48 MS/MS datasets.
- The number of MS/MS datasets selected: 1, CPTAC_TCGA_Colon_Cancer_Proteome_PDC000111
- Searching MS/MS dataset: CPTAC_TCGA_Colon_Cancer_Proteome_PDC000111. 0 left, 0 finished.
- Task type: novel peptide identification
- Unmarshaller Initialized
- All modifications in unimod:1375
- Start analysis
#####
PepQuery parameter:
- Use enzyme:Trypsin
PepQuery version: 2.0.2
PepQuery command line: -cpu 0 -s 1 -db database/gencode_v34_human.fasta -b CPTAC_TCGA_Colon_Cancer_Proteome_PDC000111 -minScore 12 -m 1 -o result/Fri_Jan_13_170743_2023_LVV/ -n 1000 -hc -plot -fast -i LVV
Fixed modification: 1 = Carbamidomethylation of C
Variable modification: 2 = Oxidation of M
Max allowed variable modification: 3
Add AA substitution: false
Enzyme: 1 = Trypsin
Max Missed cleavages: 1
Precursor mass tolerance: 20.0
Range of allowed isotope peak errors: 0
Precursor ion mass tolerance unit: ppm
Fragment ion mass tolerance: 0.6
Fragment ion mass tolerance unit: Da
Scoring algorithm: 1 = Hyperscore
Min score: 12.0
Min peaks: 10
Min peptide length: 7
Max peptide length: 45
Min peptide mass: 500.0
Max peptide mass: 10000.0
Random peptide number: 1000
Fast mode: true
CPU: 16
#####
- Spectrum ID type:1, use 1-based number as index for a spectrum.
- Step 1: target peptide sequence preparation and initial filtering ...
- Input peptide sequences:1
- Don't find indexed database:database/gencode_v34_human.fasta.sqldb
- Use database:database/gencode_v34_human.fasta
- Load db file: database/gencode_v34_human.fasta
- Start indexing fasta file
- Indexing took 0.50008991 seconds and consumes 76.949512 MB
- Ignore peptide (reason: exist in reference database): LVV
- No valid peptide!
```

3- The paper must describe the implementation in sufficient detail so that the reader can understand what is happening without having to read the other PepQuery papers. In Particular, the authors need to detail how spectra are annotated, how scores are computed, and what thresholds are used.

Response: Since spectra annotation, peptide spectrum matching algorithm as well as p-value calculation are all the same as the previous version of PepQuery (*Wen B, Wang X, Zhang B, Genome research, 2019, 29(3): 485-493*), we thought it would be redundant to repeat the same description again in the current manuscript. Based on this useful suggestion, we have added a brief description to Methods under the PSM scoring subsection in the revised manuscript to help readers understand the implementation.

Page 15: *“PSM scoring PepQuery2 uses the same peptide spectrum match (PSM) scoring and statistical evaluation algorithms as described in the original PepQuery publication¹⁶. In brief, two PSM scoring algorithms, Hyperscore³⁶ and MVH³⁷, were implemented. For statistical evaluation of each PSM, randomly shuffled sequences derived from the peptide in the PSM are used to evaluate the statistical significance of the match. The default threshold for p-value filtering is 0.01.”*

The authors also need to describe the output of the tool and the content of the web interface.

Response: The output of the tool and the content of the web interface are described in detail in our documentation available at <http://pepquery.org/document.html>.

4- A central component of the study is the validation of the peptides and PSMs. The authors mention implementing a novel validation approach, partitioning the PSMs into seven categories, one of which including “PSM passing PepQuery2 validation.” But no information is provided on what the validation refers to. The authors need to detail how validation is done, and provide some information to the reader on how to interpret these categories.

Response: Sorry about the confusion. Passing PepQuery2 validation simply means passing all filtering steps shown in Figure 1. Because the validation/filtering process is consistent with the original publication, we did not provide detailed information in this paper. During the revision, we have included more detailed information in Figure 1 legend as well as in the “Support for PSM validation” subsection in Methods (**page 14-15**) to help readers understand the process without having to go back to the original paper. In the revised manuscript, we have also reworded the sentence “*PSM passing PepQuery2 validation*” to “*PSM passing all the filtering steps as shown in Figure 1*” to avoid potential confusion.

5- The manuscript makes a strong point of examples where they were able to identify peptides that were a “better match” than the ones reported in publications. But the match being better is highly subjective of the score and spectrum annotation used. The examples presented in the manuscript are very clear. But often the difference between two matches is subtle – when does it become significant? It is not clear how the authors draw the line to consider that a match is better, this is extremely important to clarify.

Response: In PepQuery2, for a spectrum matched to a target peptide, such as a novel peptide, if there is a different peptide from the reference database matched to the spectrum with an equal or higher score than the match to the target peptide, the spectrum matched to the target peptide is considered as an unconfident match even if the two peptides have equal scores to the spectrum. We prefer to use this conservative criterium in the validation of novel or unexpected peptide identifications because the prior probability of observing novel or unexpected peptides is much lower than the competing reference peptides. These identifications should at least require more careful investigation (see manual inspection below).

In contrary, when the match to the target peptide is better than any peptides in the reference database and p-value is less than 0.01, this is considered as a putative confident match even if the score of the target peptide match is close to the best match from the reference database. The putative match is further evaluated using unrestricted modification searching, in which about 1000

modifications are considered and many peptide forms from reference database with different modifications are competed with the target peptide. This further enhances the reliability of the target peptide identification, and we have previously demonstrated its utility in reducing false positives in variant peptide identification (*Wen B, Li K, Zhang Y, et al., Nature communications, 2020, 11(1): 1-14.*).

For the examples provided in the manuscript, we provided extra evidence to support the PepQuery2 results. Specifically, for the W2F peptide validation, we use a metric derived from retention time prediction to evaluate the quality of the results. For known protein validation, we make use of paired mRNA data to support PepQuery2 results for the PGK2 and NAA11 cases. For the CHD5 example, we use prior knowledge about protein complex.

In addition, when two peptides have close scores to the same spectrum, it is very useful to manually check the matches. In PepQuery2, we export all details required for manual checking. The web server of PepQuery2 (<http://pepquery2.pepquery.org/>) provides annotated spectra for best matches from the competitive filtering based on reference sequences and competitive filtering based on unrestricted post-translational modification searching for visual check of these matches. Details can be found at <http://pepquery.org/document.html#webresult>. For the standalone version of PepQuery2, the result can be imported PDV (<https://github.com/wenbostar/PDV>) for visualization. With PDV, each spectrum matched to a target peptide can be visualized by comparing to any reference peptides matched by PepQuery2. Detailed information can be found at <http://pepquery.org/document.html#savis>.

These have been clarified in Discussion in the revised manuscript.

Page 11-12: *“Both PepQuery and PepQuery2 use stringent criteria for novel peptide validation. When a candidate novel peptide and a reference peptide have equal matching score to a spectrum, the spectrum is preferentially associated with the reference peptide based on the consideration that the prior probability of observing a novel peptide is much lower than a reference peptide. The competitive filtering step based on unrestricted modification searching further excludes the possibility that the spectrum has a better match to a reference peptide sequence with a modification not considered in the spectrum-centric analysis. When two peptides have equal or close scores to the same spectrum, it is useful to manually check the matches. The web server of PepQuery2 provides annotated spectra for visual check of the matches. For the standalone version, details required for manual checking can be exported for visualization using PDV³⁵. In this study, we also identified independent evidence to support PepQuery2 results. For the W2F peptide validation, we used a metric derived from retention time prediction to evaluate the quality of the results. For the cases of PGK2 and NAA11, we made use of paired mRNA data to support PepQuery2 results. For CHD5, we used prior knowledge about protein complex. Such analyses are very helpful in evaluating matches with equal or similar scores.”*

6- The authors have implemented different scores, and when two peptides present similar scores, they refer to the precursor mass deviation. How are these different features taken into account in the validation? Implementing something like Percolator would strengthen the approach.

Response: In PepQuery2, precursor mass deviation is only used to retrieve spectra matched to a target peptide under a specified precursor mass tolerance. It is not used in peptide spectrum match

scoring. In the example shown in Supplementary Figure 3, we did use peptide mass tolerance to support CHD4 as a preferred match to the spectrum (-0.76 ppm vs 6.85 ppm for CHD5), but this was only used as independent supporting evidence after the analysis.

It is a great idea to incorporate this information as a new feature into PSM scoring, but this cannot be easily achieved by modifying the scoring algorithms (hyperscore and MVH) used in PepQuery2. Percolator is a possible option, but it typically requires at least a few thousands PSMs from both target peptides and decoy peptides for reliable model training. The output from a typical PepQuery doesn't meet that requirement. We will consider developing new scoring schemes in PepQuery2 to support additional features in the future development.

Minor points

1- Please briefly explain c1-7 categories and how to interpret them before discussing them.

Response: Great suggestion. A brief description is added in the revised manuscript as shown below.

Page 5: *“This new feature allows users to use the peptide-centric analysis as a complementary approach to validate interesting PSMs identified in spectrum-centric analysis and classify them into seven different categories (c1-c7, **Figure 1, Methods**). C1 (exact ref match) includes input PSMs for which the peptide has an exact match to a sequence in the reference database used by PepQuery2. This is only applicable to novel peptide validation, and PSMs in this category are essentially invalid novel identifications. This could happen when the input PSMs are identified using a different database (e.g., less inclusive) than the selected reference database in PepQuery2. C2 (no candidate spectrum) includes input PSMs for which the peptide has no candidate spectrum based on the peptide mass and allowed mass error tolerance. C3 (low score) includes input PSMs with a PepQuery2 computed PSM score lower than the prespecified threshold (**Methods**). These low-quality matches are excluded from further analysis to save time. C4 (equal or better ref match) includes input PSMs for which the spectrum can be matched to a reference peptide with an equal or better PSM score. C5 (insignificant score) includes input PSMs failing the statistical evaluation based on randomly shuffled peptides (**Methods**). C6 (better mod ref match) includes input PSMs for which the spectrum can be matched to a reference peptide with a modification that is not considered in the spectrum-centric analysis. Input PSMs passing all these filtering steps are considered as C7, or confident identifications.”*

2- Please rephrase “is hard to distinguish from the KRASG12D peptide in mass spectrum.”

Response: Thanks for the suggestion. In the revised manuscript, we have rephrased this as *“the KRAS G13D peptide (LVVVGAGDVGK) is hard to distinguish from the KRAS G12D peptide (LVVVGADGVGK) based on MS/MS spectrum since there is only a minor difference between the two sequences”* (**Page 6**).

REVIEWERS' COMMENTS

Reviewer #1 (Remarks to the Author):

The authors have adequately responded to my concerns. Thank you.

Reviewer #2 (Remarks to the Author):

The authors have substantially improved their manuscript and addressed all my concerns. I recommend accepting the manuscript.

Reviewer #3 (Remarks to the Author):

I thank the authors for the detailed answers to my comments and for clarifying the text accordingly. Congratulations for this nice piece of work, we have already started the tool in our lab and it has proven very useful.

Re: NCOMMS-22-29704A “PepQuery2 democratizes public MS proteomics data for rapid peptide searching”

REVISIONS IN RESPONSE TO REVIEWERS’ COMMENTS

Reviewer #1 (Remarks to the Author):

The authors have adequately responded to my concerns. Thank you.

Response: Thank you for your insightful comments that helped improve our manuscript.

Reviewer #2 (Remarks to the Author):

The authors have substantially improved their manuscript and addressed all my concerns. I recommend accepting the manuscript.

Response: Thank you for your insightful comments that helped improve our manuscript.

Reviewer #3 (Remarks to the Author):

I thank the authors for the detailed answers to my comments and for clarifying the text accordingly. Congratulations for this nice piece of work, we have already started the tool in our lab and it has proven very useful.

Response: Thank you for your insightful comments that helped improve our manuscript. We are thrilled to know that your lab has started using the tool!